# Vicinal Label Supervision for Reliable Aleatoric and Epistemic Uncertainty Estimation

**Linye Li**
School of Computer Science and Technology
Tongji University, Shanghai, China
`linyeli@tongji.edu.cn`

**Yufei Chen***
School of Computer Science and Technology
Tongji University, Shanghai, China
`yufeichen@tongji.edu.cn`

**Xiaodong Yue**
Artificial Intelligence Institute
Shanghai University, Shanghai, China
`yswantfly@shu.edu.cn`

## Abstract

Uncertainty estimation is crucial for ensuring the reliability of machine learning models in safety-critical applications. Evidential Deep Learning (EDL) offers a principled framework by modeling predictive uncertainty through Dirichlet distributions over class probabilities. However, existing EDL methods predominantly rely on level-0 hard labels, which supervise an uncertainty-aware model with full certainty. We argue that hard labels not only fail to capture epistemic uncertainty but also obscure the aleatoric uncertainty arising from inherent data noise and label ambiguity. As a result, EDL models often produce degenerate Dirichlet distributions that collapse to near-deterministic outputs. To overcome these limitations, we propose a *vicinal risk minimization* paradigm for EDL by incorporating level-1 supervision in the form of vicinally smoothed conditional label distributions. This richer supervision exposes the model to local label uncertainty, enhancing aleatoric uncertainty quantification while mitigating the degeneration of the Dirichlet distribution into a Dirac delta function, thereby improving epistemic uncertainty modeling. Extensive experiments show that our approach consistently outperforms standard EDL baselines across synthetic datasets, covariate-shifted out-of-distribution generalization tasks, out-of-distribution detection, and selective classification benchmarks, providing more reliable uncertainty estimates.

## 1 Introduction

Reliable uncertainty estimation is pivotal for deploying trustworthy machine learning systems, particularly when models encounter distributional shifts or operate in safety-critical environments. Classical approaches, such as MC Dropout [14], Deep Ensembles [26], and Bayesian Neural Networks [4] estimate uncertainty via Bayesian model averaging. While effective, these methods often incur high computational costs due to multiple forward passes or complex posterior approximations.

Evidential Deep Learning (EDL) [45, 36, 37, 6, 7, 53, 10] has recently emerged as a promising alternative. EDL models predictive uncertainty by representing class probabilities with a Dirichlet distribution, enabling uncertainty estimation in a single forward pass without relying on sampling or model ensembles. This framework enables joint quantification of two complementary types of uncertainty within a single forward pass: (i) *Aleatoric uncertainty*, which captures inherent data noise

---

*Corresponding author.

39th Conference on Neural Information Processing Systems (NeurIPS 2025).

or label ambiguity, and (ii) *Epistemic uncertainty*, which reflects uncertainty stemming from limited data or insufficient knowledge about the data-generating process. EDL has also achieved substantial progress in several downstream tasks, such as trusted multi-view classification [15, 31, 32, 34, 49, 33, 29] and domain adaption [42, 8, 35, 54], where robust and reliable uncertainty estimation is essential.

However, since ground-truth Dirichlet distributions are unavailable, most EDL methods still rely on one-hot labels (i.e., level-0 supervision), similar to conventional softmax classifiers. Recent studies [2, 3, 22, 46] have revealed fundamental limitations of this practice: training with hard labels drives the predictive distribution to collapse into degenerate Dirac delta functions, resulting in overconfident outputs and poor uncertainty estimation. To address this issue, follow-up work [20] introduced level-1 supervision using crowdsourced soft labels (e.g., CIFAR-10H [43]), but such approaches require extensive human annotation (up to 500k judgments), making them costly and impractical. Moreover, existing studies mainly focus on epistemic uncertainty under level-0 supervision, while its ability to capture aleatoric uncertainty remains largely unexplored.

In this paper, we argue that the mismatch between fully certain level-0 supervision and uncertainty-aware level-2 predictions fundamentally limits accurate estimation of epistemic and aleatoric uncertainty. Hard labels fail to capture class ambiguity and data noise, leading Dirichlet distributions to collapse into Dirac deltas and produce overconfident, unreliable estimates. To mitigate this issue, we propose a new EDL training paradigm that narrows the supervision gap by replacing hard labels with estimated level-1 conditional categorical distributions. Specifically, we adopt a vicinal risk minimization (VRM)-based strategy that constructs level-1 supervision from local feature neighborhoods, inducing continuous label distributions and improving both aleatoric and epistemic uncertainty estimation without extra annotations.

Our main contributions are as follows:

- We introduce a novel EDL training paradigm leveraging level-1 supervision via VRM to better capture aleatoric and epistemic uncertainty without additional annotation cost.

- We provide theoretical insights into how this supervision improves generalization, supported by a risk-based analysis, and enhances aleatoric and epistemic uncertainty estimation by mitigating the Dirichlet distribution's collapse towards a Dirac delta measure.

- We empirically demonstrate the effectiveness of our method in uncertainty estimation and robustness under covariate-shifted out-of-distribution generalization, selective classification, and out-of-distribution detection.

## 2 Problem Formulation

In this section, we first discuss the key differences and connections between EDL and traditional point-estimate classifiers, such as softmax-based ones. Then, we highlight the limitations of EDL, particularly its challenges in estimating epistemic uncertainty and aleatoric uncertainty.

**Basic Notations.** In a standard supervised $K$ classification setting with instance space $\mathcal{X}$, label space $\mathcal{Y} = \{y_1, \ldots, y_K\}$, and training dataset $\mathcal{D} = \{(\boldsymbol{x}^{(1)}, y^{(1)}), \ldots, (\boldsymbol{x}^{(n)}, y^{(n)})\} \subset \mathcal{X} \times \mathcal{Y}$. Following classical settings, we also assume that the data are generated i.i.d. according to an underlying joint probability $P$ over $\mathcal{X} \times \mathcal{Y}$. Correspondingly, each instance $\boldsymbol{x} \in \mathcal{X}$ is associated with a conditional distribution $p(\cdot|\boldsymbol{x})$ on $\mathcal{Y}$, such that $p(y|\boldsymbol{x})$ is the probability to observe label $y$ as an outcome given $\boldsymbol{x}$. Let $\mathbb{P}_1(\mathcal{Y})$ denote the set of probability distributions over $\mathcal{Y}$, and $\mathbb{P}_2(\mathcal{Y})$ the set of distributions over $\mathbb{P}_1(\mathcal{Y})$. Elements of $\mathbb{P}_1(\mathcal{Y})$ are called level-1 distributions, and those of $\mathbb{P}_2(\mathcal{Y})$ level-2 distributions. As in [2], different levels of distributions in classification tasks can be organized as follows:

- **Level-0 (hard labels)**: A deterministic class label $y \in \{1, \ldots, K\}$, which implicitly assumes that the sample belongs to a single class with absolute certainty.

- **Level-1 (categorical distribution)**: A probability vector $\boldsymbol{p} \in \Delta^{K-1}$ over classes, where $\Delta^{K-1}$ is the $(K-1)$-simplex. This representation captures *aleatoric uncertainty* by modeling ambiguity or inherent noise in class membership.

- **Level-2 (Dirichlet distribution)**: A second-order distribution $\text{Dir}(\boldsymbol{\alpha})$ over categorical distributions $\boldsymbol{p}$. This signal encodes both *aleatoric* and *epistemic uncertainty*. The expected value $\mathbb{E}[\boldsymbol{p}] = \boldsymbol{\alpha}/S$ (where $S = \sum_k \alpha_k$) reflects class probabilities (aleatoric), while the

concentration $S$ controls the dispersion around the mean. Low $S$ indicates high epistemic uncertainty due to limited knowledge, and high $S$ corresponds to high confidence.

In short, level-0 distribution provides no information about uncertainty, level-1 distribution can only express aleatoric uncertainty, whereas level-2 distribution can jointly represent both aleatoric and epistemic uncertainties in a principled manner.

## 2.1 Learning Predictive Level-1 Models

Given a $K$-class classification task with input space $\mathcal{X}$ and label space $\mathcal{Y}$, the model outputs a probability vector $\boldsymbol{p} = (p_1, \ldots, p_K) \in \mathbb{P}_1(\mathcal{Y})$; here, $\mathbb{P}_1(\mathcal{Y})$ denotes the probability simplex over the label space and $\mathbb{P}_1(\mathcal{Y}) = \left\{ \boldsymbol{p} \in [0,1]^K \mid \sum_{j=1}^K p_j = 1 \right\}$. The loss function for level-1 predictors takes the form

$$L_1 : \Delta^{K-1} \times \mathcal{Y} \to \mathbb{R}. \tag{1}$$

Commonly used loss functions for level-1 predictors include the cross-entropy (CE) loss and the Brier score as

$$L_1^{\mathrm{CE}}(\boldsymbol{p}, y) = -\sum_{j=1}^K \mathbb{1}_{(j=y)} \log(p_j), \quad L_1^{\mathrm{Brier}}(\boldsymbol{p}, y) = \sum_{j=1}^K (p_j - \mathbb{1}_{(j=y)})^2. \tag{2}$$

Here, $\mathbb{1}_{(j=y)}$ is the indicator function that equals 1 if class index $j$ corresponds to the true label $y$, and 0 otherwise. Let a *hypothesis space* $\mathcal{H}_1 \subset \mathbb{P}_1(\mathcal{Y})^{\mathcal{X}} = \{h : \mathcal{X} \to \mathbb{P}_1(\mathcal{Y})\}$ to be given. In traditional supervised learning, the goal is to find a hypothesis $h \in \mathcal{H}_1$ that minimizes the risk

$$R(h) := \int_{\mathcal{X} \times \mathcal{Y}} L_1(h(\boldsymbol{x}), y) \mathrm{d}P(\boldsymbol{x}, y), \tag{3}$$

where $R(h)$ is the risk of hypothesis $h$. The hypothesis is commonly learned via Empirical Risk Minimisation (ERM), which involves minimizing the empirical risk defined as:

$$\hat{R}_{\mathrm{emp}}(h; \mathcal{D}) := \frac{1}{N} \sum_{n=1}^N L_1(h(\boldsymbol{x}^{(n)}), y^{(n)}). \tag{4}$$

Since $\hat{R}_{\mathrm{emp}}(h; \mathcal{D})$ is an estimate of the true risk $R(h)$, the learned hypothesis $\hat{h}$ is an approximation of the true risk minimizer $h^*$. They are defined as follows:

$$\hat{h} := \arg \min_{h \in \mathcal{H}_1} \hat{R}_{\mathrm{emp}}(h; \mathcal{D}), \qquad h^* := \arg \min_{h \in \mathcal{H}_1} R(h). \tag{5}$$

Consequently, there remains an approximation gap between $\hat{h}$ and $h^*$, as well as epistemic uncertainty regarding $h^*$, as only a single point estimate of the predictive distribution is obtained [14, 45].

## 2.2 Learning Predictive Level-2 Models

Unlike level-1 models, EDL learns a hypothesis space $\mathcal{H}_2$ of the form $\mathcal{H}_2 \subset \mathbb{P}_2(\mathcal{Y})^{\mathcal{X}} = \{h : \mathcal{X} \to \mathbb{P}_2(\mathcal{Y})\}$. To learn a level-2 predictor, the ideal scenario would involve access to a ground-truth distribution $Q^* \in \mathbb{P}_2(\mathcal{Y})$, which could directly supervise the model to output the target level-2 distribution. However, such ground-truth distributions $Q^*$ are typically unavailable in practice. Consequently, existing methods adopt an alternative approach inspired by level-1 models. Specifically, a level-2 loss function is defined as

$$L_2 : \mathbb{P}_2(\mathcal{Y}) \times \mathcal{Y} \to \mathbb{R}_+, \tag{6}$$

which compares the level-2 prediction $h(\boldsymbol{x})$ against a level-0 observation $y$. The learning objective is to minimize the empirical level-2 risk over the training data $\mathcal{D}$ as

$$\hat{R}_{\mathrm{emp}}^{(2)}(h) = \frac{1}{N} \sum_{n=1}^N L_2(h(\boldsymbol{x}^{(n)}), y^{(n)}). \tag{7}$$

This paradigm, known as *evidential deep learning* (EDL), aims to learn a reliable level-2 distribution predictor for uncertainty estimation by minimizing the $L_2$ loss on the available data [45, 36, 6]. Several previous works have proposed the minimization of an empirical loss of the form

$$L_2(Q, y) = \mathbb{E}_{\boldsymbol{p} \sim Q} L_1(\boldsymbol{p}, y), \tag{8}$$

where the level-2 prediction $Q$ is penalized in terms of the *expected* level-1 loss.

## 2.3 Limitations of Aleatoric and Epistemic Uncertainty Estimation in EDL

However, recent studies have raised substantial criticisms regarding the uncertainty estimation behavior of EDL. Specifically, it has been argued that minimizing empirical risk under standard EDL frameworks with level-0 observations inevitably drives the learned evidential distribution to collapse into a Dirac measure. Consequently, epistemic uncertainty is effectively suppressed or unreported in practice [23, 22, 3, 2]. Building upon these insights, we conduct a systematic analysis of epistemic uncertainty estimation in EDL and rigorously formalize how the distributional collapse phenomenon undermines its ability to quantify uncertainty. Furthermore, we identify and theoretically characterize an additional, underexplored limitation: EDL also fails to faithfully capture aleatoric uncertainty under the same empirical risk minimization principle. Specifically, we establish the following result:

**Theorem 1.** *For any level-1 loss function $L_1 : \mathbb{P}_1(\mathcal{Y}) \times \mathcal{Y} \to \mathbb{R}$ that satisfies $L_1(\mathbb{E}_{p \sim Dir(\alpha)} p, \cdot) \leq \mathbb{E}_{p \sim Dir(\alpha)} L_1(p, \cdot)$, (i.e., is a convex function), such as Brier score and the log-loss in Eq. 2, the empirical risk minimizer of a level-2 prediction is always a Dirac measure $\delta_p \in \mathbb{P}_2(\mathcal{Y})$ and the expectation of level-2 prediction is $\delta_y \in \mathbb{P}_1(\mathcal{Y})$. This result holds if the learner possesses a universal approximation property, allowing it to represent such a degenerate distribution.*

Here, $\delta_y$ denotes the Dirac measure at $y \in \mathcal{Y}$, which is an element of $\mathbb{P}_1(\mathcal{Y})$ representing a prediction with no aleatoric uncertainty. While prior work [2, 3, 22] has established that Empirical Risk Minimization (ERM) with level-0 labels leads to a collapse of epistemic uncertainty for second-order predictors like EDL, Theorem 1 highlights that aleatoric uncertainty also vanishes under the same conditions. This observation is inspired by the analysis in Theorem 3.3 of [22], which shows that the $L_1$ loss is minimized when the prediction is a Dirac measure $\delta_y$ centered on the ground-truth label $y$. The complete proof is provided in the Appendix B. An intuitive explanation is that under level-0 supervision, even if the predicted distribution $p$ accurately reflects the learner's aleatoric uncertainty, it is not the minimizer of Eq. 8. As a result, a learner trained to minimize this loss with level-0 labels will not output such a distribution $p$. Instead, the optimal prediction becomes the mode of $p$, leading to a degenerate distribution $\delta_y$ that collapses the uncertainty representation into a single point mass.

**Proposition 1.** *Under the assumptions of Theorem 1, empirical risk minimization of level-2 prediction inevitably yields degenerate distributions $\delta_p \in \mathbb{P}_2(\mathcal{Y})$ and the expectation of the level-2 prediction is $\delta_y \in \mathbb{P}_1(\mathcal{Y})$. As a result, the model fails to provide any meaningful or disentangled representation of aleatoric or epistemic uncertainty.*

This proposition follows directly from Theorem 1: since the optimal prediction always collapses to a Dirac measure at the expected probability vector, the learner is incentivized to output deterministic posteriors, irrespective of whether the uncertainty arises from stochastic labels (aleatoric) or from limited evidence (epistemic). Consequently, any observed uncertainty in the model's prediction cannot be separated into aleatoric and epistemic components.

As level-0 labels provide fully certain supervision, learning aleatoric uncertainty over $p$ remains challenging. A natural solution is to leverage level-1 soft labels, as in [20]. However, most standard datasets [24, 13] only offer hard labels, and collecting accurate level-1 annotations (e.g., CIFAR10-H [43]) is impractical. To address this, we propose a VRM strategy that approximates soft labels by interpolating between neighboring samples.

## 3 Method

Our proposed method enhances uncertainty estimation in Dirichlet-based models by leveraging VRM-approximated soft labels. This strategy enables learning aleatoric uncertainty from datasets restricted to hard labels. The method consists of two complementary components: one strengthens the model's capacity for aleatoric uncertainty capture, while the other preserves epistemic uncertainty by preventing the Dirichlet distribution from degenerating into a Dirac delta function.

### 3.1 Vicinal Supervision to Enhance Aleatoric Uncertainty Estimation

The empirical risk minimization trains a model to minimize loss on the exact training samples $(x^{(n)}, y^{(n)})$, but ignores the fact that the true data distribution $P(x, y)$ is continuous and often smooth in the instance-label space $\mathcal{X} \times \mathcal{Y}$. VRM [5] addresses the continuity of the instance space by introducing a vicinal distribution, which augments the training set with virtual examples drawn

from local neighborhoods of the data while keeping the labels unchanged. The Mixup method [51] extends the VRM principle by applying linear interpolations not only to the input features but also to the labels. The empirical vicinal risk in this manner is defined as:

$$\hat{R}_v(h; \mathcal{D}) := \frac{1}{N} \sum_{n=1}^{N} \int \int L(h(\tilde{\boldsymbol{x}}), \tilde{\boldsymbol{y}}) \, p(\tilde{\boldsymbol{x}}, \tilde{\boldsymbol{y}} | \boldsymbol{x}^{(n)}, \boldsymbol{y}^{(n)}) \, \mathrm{d}\tilde{\boldsymbol{x}} \, \mathrm{d}\tilde{\boldsymbol{y}}, \tag{9}$$

where the vector $\boldsymbol{y}$ denotes the label in one-hot format and $p(\tilde{\boldsymbol{x}}, \tilde{\boldsymbol{y}} | \boldsymbol{x}^{(n)}, \boldsymbol{y}^{(n)})$ represents the joint probability density function of vicinal samples. In practice, the interpolated samples and labels are generated by linearly interpolating between pairs of training examples:

$$\tilde{\boldsymbol{x}} = \lambda \boldsymbol{x}^{(n)} + (1 - \lambda) \boldsymbol{x}^{(m)}, \qquad \tilde{\boldsymbol{y}} = \lambda \boldsymbol{y}^{(n)} + (1 - \lambda) \boldsymbol{y}^{(m)}, \tag{10}$$

where $(\boldsymbol{x}^{(m)}, \boldsymbol{y}^{(m)})$ is another randomly selected training sample, and $\lambda \in [0, 1]$ is drawn from a Beta distribution $\mathrm{Beta}(\beta, \beta)$ with a hyperparameter $\beta > 0$. The hyperparameter $\beta$ controls the shape of the Beta distribution. When $\beta < 1$, the distribution is U-shaped, favoring extreme values of $\lambda \approx 0$ or $\lambda \approx 1$, which leads to mixtures dominated by a single sample. In contrast, when $\beta \gg 1$, the distribution concentrates around $\lambda \approx 0.5$, promoting strongly balanced mixtures between the two samples. Inspired by the idea of Vicinal Risk Minimization (VRM), we incorporate vicinal information by generating vicinal level-1 labels $\tilde{\boldsymbol{y}}$, which are subsequently used as the supervision target of the $L_1$ loss. The resulting objective is formulated as:

$$\mathcal{L}_{\text{vicinal}} = \mathbb{E}_{(\boldsymbol{x}^{(n)}, \boldsymbol{y}^{(n)}), (\boldsymbol{x}^{(m)}, \boldsymbol{y}^{(m)}) \sim \mathcal{D}} \mathbb{E}_{\lambda \sim \mathrm{Beta}(\beta, \beta)} \mathbb{E}_{\boldsymbol{p} \sim \mathrm{Dir}(\tilde{\boldsymbol{\alpha}} | \tilde{\boldsymbol{x}})} \left[ L_1(\boldsymbol{p}, \tilde{\boldsymbol{y}}) \right]. \tag{11}$$

In contrast to the original Mixup [51], which suggests setting $\beta = 0.2$ or $0.4$, we set $\beta \gg 1$ (e.g., 10, 20) to enforce strong mixing. Such strong mixing creates soft labels that represent high aleatoric uncertainty, encouraging the model to account for inherent label ambiguity and improve uncertainty calibration.

## 3.2 Noise-Augmented Vicinal Risk Minimization for Epistemic Uncertainty Estimation

In addition to simulating samples with high aleatoric uncertainty, we introduce controlled noise into the input space to generate samples that exhibit inherently high epistemic uncertainty—i.e., samples that are difficult to model due to insufficient, ambiguous, or incomplete information [19]. Specifically, we propose augmenting VRM with synthetic vague samples generated via Gaussian noise to account for the simple fact that the observed features $\boldsymbol{x}$ may not contain sufficient information to fully explain the target $\boldsymbol{y}$. We formalize this with the following noise-augmented loss:

$$\mathcal{L}_{\text{noise}} = \mathbb{E}_{(\boldsymbol{x}^{(n)}, \boldsymbol{y}^{(n)}) \sim \mathcal{D}, \boldsymbol{x}^{(m)} \sim \mathcal{N}(0, \sigma^2 I)} \mathbb{E}_{\lambda \sim \mathrm{Beta}(\beta_{\text{noise}}^+, \beta_{\text{noise}}^-)} \mathbb{E}_{\boldsymbol{p} \sim \mathrm{Dir}(\tilde{\boldsymbol{\alpha}} | \tilde{\boldsymbol{x}})} \left[ L_1(\boldsymbol{p}, \tilde{\boldsymbol{y}}) \right]. \tag{12}$$

Here, $\boldsymbol{x}^{(m)}$ is sampled from Gaussian noise, label $\boldsymbol{y}^{(m)}$ is set as a uniform distribution

$$\tilde{\boldsymbol{x}} = \lambda \boldsymbol{x}^{(n)} + (1 - \lambda) \boldsymbol{x}^{(m)}, \qquad \tilde{\boldsymbol{y}} = \lambda \boldsymbol{y}^{(n)} + (1 - \lambda) \left[ \frac{1}{K}, ..., \frac{1}{K} \right]. \tag{13}$$

While the added noise can inadvertently increase aleatoric uncertainty, we primarily introduce it to simulate obstacles to the model's knowledge acquisition and encourages the model to produce smoother uncertainty estimates in the vicinities of the training data. Furthermore, we observe that adding label smoothing helps control the growth of the Dirichlet strength for samples near the decision boundary, mitigating the degeneration of the Dirichlet distribution into a Dirac delta function, as formalized in Theorem 3. The Beta distribution parameters $\beta_{\text{noise}}^+$ and $\beta_{\text{noise}}^-$ govern the mixing proportion. A larger $\beta_{\text{noise}}^-$ increases the contribution of noisy samples (i.e., smaller $\lambda$ on average); A larger $\beta_{\text{noise}}^+$ emphasizes the clean (original) samples. We set $\beta_{\text{noise}}^+ \geq \beta_{\text{noise}}^-$ to ensure that original samples dominate the interpolation, thereby preventing excessive degradation of predictive performance while still allowing the model to explore uncertain vicinities.

## 3.3 Model optimization

We follow EDL [45] to train a neural network that predicts the parameters of a Dirichlet distribution. Specifically, we set the Dirichlet prior as a non-informative prior $\boldsymbol{a} = [1, \ldots, 1]$. The neural network's outputs $\Phi(\boldsymbol{x})$ are passed through a non-negative activation function $\sigma(\cdot)$, e.g. ReLU

or SoftPlus, to obtain the evidence vector $\boldsymbol{e} = \{e_1, \ldots, e_K\}$, i.e., $\boldsymbol{e} = \sigma(\Phi(\boldsymbol{x}))$. The Dirichlet parameters are then computed as $\boldsymbol{\alpha} = \boldsymbol{e} + \boldsymbol{a}$. For the loss function, we adopt the cross-entropy-based EDL loss, defined as:

$$
\begin{aligned}
\mathbb{E}_{\boldsymbol{p} \sim \mathrm{Dir}(\tilde{\boldsymbol{\alpha}})}[L_1(\boldsymbol{p}, \tilde{\boldsymbol{y}})] &= \int \left[ \sum_{j=1}^{K} -\tilde{y}_j \log(p_j) \right] \frac{1}{\mathrm{B}(\boldsymbol{\alpha})} \prod_{j=1}^{K} p_j^{\alpha_j - 1} d\boldsymbol{p} \\
&= \sum_{j=1}^{K} \tilde{y}_j \big( \psi(S) - \psi(\alpha_j) \big),
\end{aligned}
\tag{14}
$$

where $S = \sum_{j=1}^{K} \alpha_j$, $\psi(\cdot)$ is the digamma function. Finally, the total vicinal loss is defined as:

$$
\mathcal{L} = \mathcal{L}_{\mathrm{vicinal}} + \mathcal{L}_{\mathrm{noise}}.
\tag{15}
$$

By jointly optimizing the total loss, the model can not only fit the training data effectively but also express predictive uncertainty more reliably and generalize better to unseen or ambiguous scenarios.

## 4 Theoretical Analysis

We provide a theoretical analysis to elucidate how level-1 labels with strong mixing improve generalization and robustness in the presence of input-dependent label noise. Then, we analyze how the hyperparameter $\lambda$ in Eq. 13 moderates the rapid increase of the Dirichlet concentration, thereby slowing its degeneration towards a Dirac delta function. The complete proof is provided in the Appendix B.

**Theorem 2.** *Let the ground-truth level-1 label be denoted as $\boldsymbol{p}^*(\boldsymbol{x})$, and let the observed level-0 one-hot label $\delta_y(\boldsymbol{x})$ be a noisy realization of $\boldsymbol{p}^*(\boldsymbol{x})$ perturbed by input-dependent label noise $\boldsymbol{\mu}(\boldsymbol{x})$*

$$
\delta_y(\boldsymbol{x}) = \boldsymbol{p}^*(\boldsymbol{x}) + \boldsymbol{\mu}(\boldsymbol{x}) \quad where \quad \boldsymbol{\mu}(\boldsymbol{x}) \sim \mathcal{N}(\boldsymbol{0}, \sigma^2 \boldsymbol{I}).
\tag{16}
$$

*Then, the test risk admits the following lower bound under mild regularity conditions*

$$
R(\hat{h}; P) \geq C\sigma^2,
\tag{17}
$$

*where $C$ depends on the trace of the Hessian matrix of the loss function with respect to $\boldsymbol{p}$. Then, for the level-1 label with strong mixing, the bound can be tightened as*

$$
R(\hat{h}; P) \geq C'\sigma^2,
\tag{18}
$$

*where $C'/C \approx \frac{1}{2\beta+1} + \frac{1}{2} < 1$ ($\forall \beta \gg 1/2$), indicating a reduced sensitivity of the test risk to input-dependent noise.*

Theorem 2 implies that leveraging level-1 labels with strong mixing can effectively reduce the lower bound of the generalization error.

**Theorem 3.** *Let $\lambda$ be the mixing hyperparameter defined in Eq. 13. Consider the optimization of the Dirichlet parameters $\boldsymbol{\alpha}$ in Eq. 14. For samples where $\alpha_k \leq \alpha_j$ ($\forall j \neq k$) with lower belief assigned to the ground-truth $k$ class, the following properties hold*

- *The update to the Dirichlet concentration for the ground-truth class $\Delta\alpha_k$, increases monotonically with $\lambda$.*

- *The updates to the Dirichlet concentrations for the non-ground-truth classes $\Delta\alpha_{j \neq k}$, decrease monotonically with $\lambda$.*

- *The total increase in Dirichlet concentration, denoted $\Delta S$, increases monotonically with $\lambda$.*

Theorem 3 implies that a properly chosen $\lambda < 1$ can effectively suppress the excessive accumulation of evidence, i.e., $\Delta S$. This prevents Dirichlet-based models from collapsing into a degenerate Dirac delta distribution $\delta_p$, thereby enhancing their ability to represent epistemic uncertainty.

# 5 Experiments

We begin by analyzing and comparing the estimated uncertainties estimated by our method and baseline approaches on a toy dataset. Subsequently, we conduct extensive experiments on three main tasks: *OOD detection*, *selective classification*, and *OOD generalization*. For the OOD detection task, we evaluate the ability of different methods to distinguish between in-distribution (ID) and out-of-distribution (OOD) samples based on their estimated *epistemic uncertainty*. For selective classification, we assess the model's capability to differentiate correctly classified samples from misclassified ones using *aleatoric uncertainty*. For the OOD generalization task, we examine the classification performance of models when exposed to covariate-shifted OOD samples.

## 5.1 Experimental Setup

**Baselines.** Baseline methods include KL-PN [36], RKL-PN [37], PostNet [6], NatPN [7], EDL [45], RED [40], $\mathcal{I}$-EDL [12], R-EDL [9], H-EDL [44], and DA-EDL [50]. For OOD detection tasks, we further extend our experiments to include four OOD detection methods based on uncertainty estimation: DUQ [47], DDU [38], DUE [48], and SNGP [30].

**Evaluation Metrics.** We evaluate OOD detection performance using the Area Under the Receiver Operating Characteristic curve (AUROC), which measures the model's ability to distinguish between ID and OOD samples. For selective classification, we employ the Excess Area Under the *Risk-Coverage Curve* (E-AURC × 1000; lower is better), where a lower E-AURC indicates more reliable aleatoric uncertainty estimation and better selective prediction performance. OOD generalization is assessed by measuring the classification accuracy on covariate-shifted OOD test sets. For comparison, the classification accuracy on the ID test set is also reported. All results are reported as the mean $\pm$ standard deviation over 10 independent runs with different random seeds.

**Implementation Details.** Following OpenOOD [52], we train a ResNet-18 model [16] implemented in PyTorch [41] for 100 epochs on a single NVIDIA A100 GPU. We use the SGD optimizer with a cosine annealing schedule, an initial learning rate of 0.1, and a batch size of 128. We set the hyperparameters $\beta = 10$ (Eq. 11) and $\beta_{\text{noise}}^{+} = \beta_{\text{noise}}^{-} = 1.0$. Further implementation details for the baselines are in Appendix D.

**Uncertainty Measure.** Existing methods adopt different strategies to quantify epistemic uncertainty. KL-PN [36] and RKL-PN [37] use mutual information (Eq. 66); PostNet [6] and NatPN [7] rely on the Dirichlet total strength $S = \sum_{j=1}^{K} \alpha_j$, where a smaller $S$ indicates higher uncertainty. Methods based on Dempster-Shafer Theory and Subjective Logic [45, 40, 12, 9, 44, 50] use vacuity ($K/S$) as their measure of uncertainty. We propose using conditional entropy for aleatoric uncertainty (Eq. 64) and the Dirichlet differential entropy for epistemic uncertainty (Eq. 67).

**Datasets.** Following prior EDL works, we conduct OOD detection using CIFAR-10 and CIFAR-100 [24] (32 × 32 resolution). When using CIFAR-10 (or CIFAR-100) as the ID dataset, the OOD datasets include CIFAR-100 (or CIFAR-10), Tiny ImageNet [27], MNIST [28], SVHN [39], Textures [25], and Places365 [55]. For OOD generalization, we evaluate on CIFAR-10-C and CIFAR-100-C [17], which contain 15 corruption types (e.g., snow, fog) at 5 severity levels.

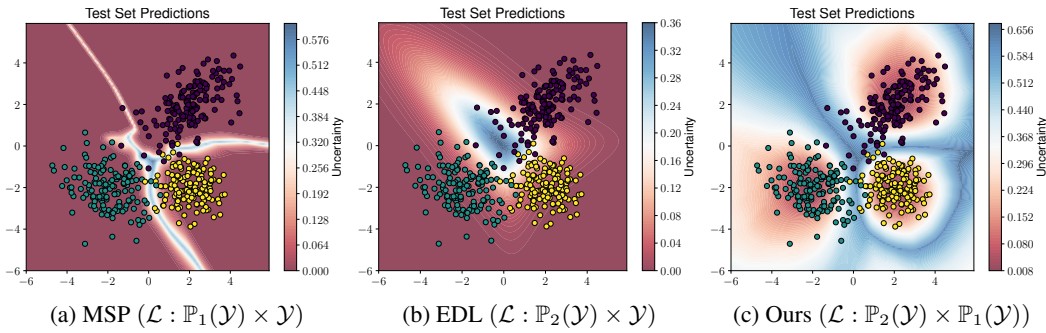

(a) MSP ($\mathcal{L} : \mathbb{P}_1(\mathcal{Y}) \times \mathcal{Y}$)    (b) EDL ($\mathcal{L} : \mathbb{P}_2(\mathcal{Y}) \times \mathcal{Y}$)    (c) Ours ($\mathcal{L} : \mathbb{P}_2(\mathcal{Y}) \times \mathbb{P}_1(\mathcal{Y})$)

Figure 1: Comparison of estimated uncertainty of different methods on toy dataset.

Table 1: OOD detection (AUROC) and classification accuracy on CIFAR-10 and CIFAR-100 dataset.

| Method | →CIFAR-100 OOD Detect | →Tiny OOD Detect | →MNIST OOD Detect | →SVHN OOD Detect | →Textures OOD Detect | →Places365 OOD Detect | CIFAR-10 Cls Acc |
|---|---|---|---|---|---|---|---|
| DUQ [ICML20] [47] | 84.60±1.04 | 86.16±0.92 | 92.34±1.25 | 91.36±1.26 | 86.57±1.27 | 84.26±0.93 | 93.60±0.39 |
| DDU [CVPR23] [38] | **89.23±0.33** | 91.28±0.28 | 95.69±0.99 | 93.53±1.43 | 92.64±0.31 | 91.55±0.34 | 95.35±0.05 |
| DUE [48] | 86.00±0.78 | 88.40±0.76 | 92.34±1.48 | 91.70±1.63 | 89.25±1.16 | 88.89±0.77 | 94.98±0.15 |
| SNGP [NIPS20] [30] | 88.14±1.18 | 90.38±1.00 | 93.18±1.20 | 92.16±2.40 | 92.88±1.37 | 91.21±1.20 | 94.63±0.18 |
| KL-PN [NIPS18] [36] | 83.46±0.88 | 85.24±0.97 | 87.80±3.81 | 88.39±2.37 | 85.41±1.36 | 84.36±0.77 | 90.31±1.46 |
| RKL-PN [NIPS19] [37] | 60.43±2.54 | 63.04±2.43 | 85.76±3.03 | 43.97±9.92 | 59.17±3.17 | 66.05±2.62 | 53.22±4.25 |
| PostNet [NIPS20] [6] | 81.46±1.00 | 77.78±3.89 | 85.19±2.19 | 88.27±1.76 | 85.87±1.23 | 82.97±1.76 | 89.44±0.62 |
| NatPN [NIPS21] [7] | 78.44±0.83 | 80.24±0.41 | 81.09±4.39 | 82.23±1.20 | 82.23±5.12 | 80.76±0.49 | 86.25±0.40 |
| EDL [NIPS18] [45] | 86.64±0.25 | 90.59±0.21 | 93.24±0.62 | 93.56±1.01 | 90.66±0.62 | 90.25±0.34 | 94.15±0.24 |
| RED [ICML23] [40] | 85.81±0.34 | 88.07±0.27 | 91.60±1.50 | 92.12±0.92 | 88.08±1.47 | 88.05±0.38 | 94.83±0.18 |
| $\mathcal{I}$-EDL [ICML23] [12] | 88.11±0.45 | 90.83±0.45 | 94.20±1.11 | 94.77±1.62 | 91.29±0.96 | 90.38±0.40 | 94.95±0.17 |
| R-EDL [ICLR24] [9] | 86.88±0.08 | 89.72±0.47 | 90.66±1.40 | 92.53±4.25 | 90.79±0.79 | 87.06±0.31 | 92.92±0.13 |
| H-EDL [NIPS24] [44] | 88.60±0.29 | 91.43±0.19 | 95.60±0.27 | 92.99±0.67 | **92.97±0.34** | **92.07±0.32** | 95.04±0.05 |
| DA-EDL [ICML24] [50] | 82.39±0.65 | 84.03±0.63 | 88.80±0.33 | 86.98±0.69 | 82.27±0.87 | 83.40±0.45 | 92.57±0.15 |
| Ours | 89.09±0.19 | **91.81±0.22** | **97.32±0.42** | **96.20±0.32** | 92.51±0.73 | 91.61±0.15 | **96.18±0.13** |

| Method | →CIFAR-10 OOD Detect | →Tiny OOD Detect | →MNIST OOD Detect | →SVHN OOD Detect | →Textures OOD Detect | →Places365 OOD Detect | CIFAR-100 Cls Acc |
|---|---|---|---|---|---|---|---|
| DUQ [ICML20] [47] | 51.20±1.84 | 53.60±2.67 | 39.44±13.20 | 61.47±7.64 | 57.73±5.66 | 50.16±3.13 | 1.66±0.39 |
| DDU [CVPR23] [38] | 68.14±1.63 | 78.64±1.57 | 79.69±6.56 | 76.02±3.98 | **83.00±1.01** | 74.53±1.70 | 78.05±0.96 |
| DUE [48] | 50.30±1.54 | 49.97±1.39 | 49.91±1.43 | 49.91±1.02 | 50.02±1.65 | 50.12±1.01 | 1.06±0.17 |
| SNGP [NIPS20] [30] | 72.77±1.23 | 76.63±1.51 | 71.91±6.19 | 73.54±5.36 | 73.91±1.85 | 74.53±1.93 | 76.19±1.19 |
| KL-PN [NIPS18] [36] | 57.20±2.79 | 60.56±1.86 | 55.20±22.34 | 50.90±12.39 | 49.38±3.61 | 57.93±2.60 | 24.89±10.48 |
| RKL-PN [NIPS19] [37] | 53.13±2.15 | 51.30±1.38 | 48.27±25.03 | 55.46±10.83 | 44.35±4.15 | 54.24±3.88 | 17.64±2.80 |
| PostNet [NIPS20] [6] | 54.19±0.59 | 53.58±0.51 | 75.93±11.81 | 59.89±8.15 | 52.08±4.59 | 53.86±1.09 | 3.08±0.23 |
| NatPN [NIPS21] [7] | 67.77±0.87 | 70.69±0.69 | 65.20±4.90 | 75.34±2.37 | 66.53±1.60 | 69.25±0.83 | 59.01±0.40 |
| EDL [NIPS18] [45] | 56.49±2.47 | 57.40±1.92 | 28.84±6.81 | 53.78±14.68 | 49.68±3.88 | 56.74±2.68 | 30.23±2.72 |
| RED [ICML23] [40] | 78.17±0.35 | 81.79±0.18 | 78.45±2.53 | 80.98±2.42 | 77.79±0.31 | 78.71±0.40 | 77.60±0.26 |
| $\mathcal{I}$-EDL [ICML23] [12] | 77.42±0.31 | 82.39±0.29 | 76.22±0.83 | 78.91±0.25 | 78.54±0.31 | 79.65±0.19 | 77.10±0.12 |
| R-EDL [ICLR24] [9] | 65.45±2.01 | 71.28±0.53 | 79.44±4.42 | 78.50±2.49 | 74.35±1.37 | 75.35±0.94 | 46.70±1.69 |
| H-EDL [NIPS24] [44] | 75.73±0.39 | 80.81±0.26 | 73.91±4.34 | 83.56±3.07 | 75.74±0.70 | 79.97±0.59 | 77.75±0.23 |
| DA-EDL [ICML24] [50] | 53.07±1.03 | 56.78±0.64 | 65.63±3.12 | 48.02±2.34 | 49.53±2.21 | 54.48±2.31 | 16.59±0.76 |
| Ours | **81.86±0.19** | **83.40±0.09** | **86.17±1.83** | **85.07±1.94** | 78.58±0.33 | **79.97±0.39** | **78.29±0.11** |

## 5.2 Experimental Results

**Uncertainty estimation on toy dataset.** We begin by conducting an experiment on a toy dataset consisting of three Gaussian clusters, as shown in Fig. 1. To maintain consistency, the uncertainty measure for all methods is defined as the vacuity of evidence, i.e., $K/S$. Obviously, with level-1 supervision, our method yields more precise uncertainty estimation; in particular, it assigns higher uncertainty to regions that are far from the in-distribution data and near decision boundaries.

**Evaluation of epistemic uncertainty via OOD detection.** As shown in Table 1, our method achieves competitive performance in detecting OOD samples. By leveraging vicinal label information, our method learns a smoother uncertainty landscape, leading to more reliable uncertainty estimates without imposing any regularization on the output Dirichlet distribution. Besides, as demonstrated in Theorem 3, our method mitigates the tendency of the Dirichlet distribution for uncertain samples to collapse towards a Dirac delta function, further enhancing the accuracy of uncertainty estimation.

**Evaluation of aleatoric uncertainty via selective classification.** As shown in Table 2, our method consistently achieves the lowest E-AURC across all corruption severities. This result demonstrates more reliable aleatoric uncertainty estimation and improved performance in selective classification scenarios. This suggests that many existing methods do not adequately model aleatoric uncertainty, particularly on corrupted data. In contrast, our method addresses this limitation by explicitly modeling the aleatoric uncertainty inherent in level-1 labels, amplified through a strong mixup strategy, thereby achieving a more robust uncertainty characterization.

**OOD generalization performance.** As shown in Table 3, our method demonstrates significantly superior OOD generalization. This addresses a critical limitation of previous EDL methods, which often suffer from a severe degradation in classification accuracy on OOD data, thereby limiting their practical applicability. Our approach overcomes this issue, as theoretically supported by Theorem 2. By employing a strong mixup strategy with $\beta \gg 1/2$, our method substantially reduces the model's sensitivity to input-dependent noise while simultaneously enhancing its generalization capabilities, leading to robust performance even on out-of-distribution inputs.

Table 2: Selective classification results on CIFAR-10-C using E-AURC at different level of severity $s$.

| Method | $s{=}1$ | $s{=}2$ | $s{=}3$ | $s{=}4$ | $s{=}5$ | Mean |
|---|---|---|---|---|---|---|
| EDL | 18.12±0.31 | 30.16±0.96 | 44.54±1.95 | 63.61±1.90 | 101.61±4.46 | 51.60±1.91 |
| RED | 16.74±0.40 | 30.15±1.40 | 44.80±1.66 | 65.26±2.61 | 103.87±5.73 | 52.16±2.36 |
| $\mathcal{I}$-EDL | 14.62±0.52 | 27.84±0.65 | 41.70±1.57 | 59.53±2.92 | 95.93±4.43 | 47.92±2.01 |
| R-EDL | 17.13±0.47 | 29.85±1.05 | 45.06±0.74 | 63.75±0.51 | 101.74±1.21 | 51.50±0.79 |
| DA-EDL | 20.61±2.75 | 35.56±5.31 | 51.09±8.38 | 72.29±11.23 | 112.83±14.53 | 58.47±8.44 |
| **Ours** | **8.70±0.35** | **14.80±0.45** | **21.90±0.81** | **35.06±1.81** | **66.52±7.37** | **29.40±2.16** |

Table 3: OOD generalization accuracy on CIFAR10-C and CIFAR100-C dataset.

| Dataset | Method | $s = 1$ | $s = 2$ | $s = 3$ | $s = 4$ | $s = 5$ | Mean |
|---|---|---|---|---|---|---|---|
| CIFAR10-C | EDL | 87.44±0.28 | 80.90±0.55 | 75.25±0.70 | 68.08±0.93 | 56.24±1.10 | 73.58±0.71 |
| | RED | 88.23±0.21 | 81.38±0.37 | 75.48±0.66 | 68.05±0.97 | 56.61±1.28 | 73.95±0.70 |
| | $\mathcal{I}$-EDL | 88.03±0.21 | 81.10±0.38 | 75.32±0.53 | 68.12±0.48 | 56.98±0.51 | 73.01±0.42 |
| | R-EDL | 85.46±0.32 | 80.12±0.43 | 74.91±0.51 | 67.53±0.68 | 56.74±0.88 | 72.95±0.56 |
| | H-EDL | 87.82±0.15 | 81.66±0.21 | 76.23±0.25 | 67.64±0.30 | 55.88±0.33 | 73.85±0.25 |
| | DA-EDL | 85.26±0.34 | 80.06±0.58 | 74.88±0.69 | 67.82±0.88 | 57.87±1.06 | 73.18±0.71 |
| | Ours | **93.88±0.09** | **92.04±0.09** | **90.19±0.12** | **87.02±0.18** | **80.53±0.15** | **88.73±0.13** |
| CIFAR100-C | EDL | 26.37±2.43 | 22.82±2.24 | 20.69±2.15 | 18.14±1.98 | 14.98±1.78 | 20.60±2.12 |
| | RED | 64.69±0.20 | 55.65±0.23 | 50.02±0.27 | 43.36±0.30 | 33.06±0.24 | 49.36±0.25 |
| | $\mathcal{I}$-EDL | 64.43±0.26 | 55.52±0.33 | 49.82±0.34 | 43.16±0.30 | 32.72±0.41 | 49.13±0.33 |
| | R-EDL | 40.42±1.98 | 35.34±1.74 | 32.13±1.59 | 28.20±1.35 | 22.71±0.97 | 31.76±1.53 |
| | H-EDL | 64.87±0.21 | 55.85±0.15 | 50.22±0.16 | 43.68±0.15 | 33.35±0.18 | 49.59±0.17 |
| | DA-EDL | 16.76±1.55 | 14.94±1.32 | 13.81±0.99 | 12.53±0.99 | 11.10±0.85 | 13.83±1.17 |
| | Ours | **69.34±0.25** | **65.29±0.32** | **63.07±0.29** | **58.64±0.37** | **50.46±0.37** | **61.35±0.32** |

**Visualization of estimated uncertainty.** In Figs. 2a and 2b, we visualize the uncertainty distributions produced by our model and a baseline method (MSP with vicinal training). These figures show that point-estimate-based methods, despite supervision from level-1 labels, exhibit limited improvement in OOD detection and remain overconfident on OOD samples. In contrast, our method achieves a clear separation between ID and OOD samples. To evaluate the model's ability to estimate aleatoric uncertainty, we further visualize its predictions on CIFAR-10 and five CIFAR-10-C variants with increasing corruption severity (Fig. 2c). As the corruption level increases, the estimated aleatoric uncertainty rises accordingly, indicating that our model reliably captures data uncertainty.

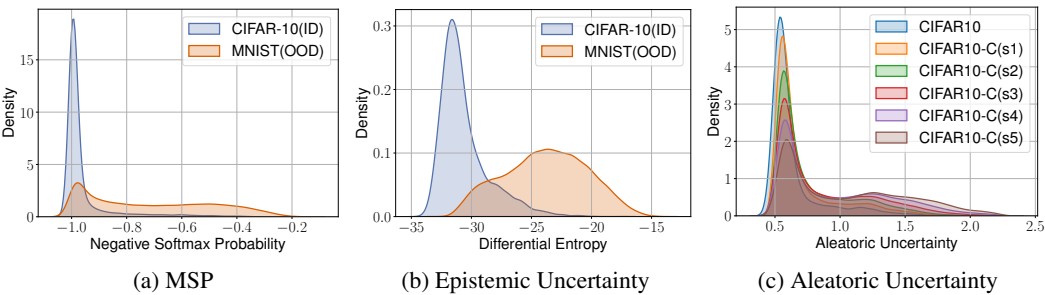

|          (a) MSP          |   (b) Epistemic Uncertainty   |   (c) Aleatoric Uncertainty   |

Figure 2: Visualization of uncertainties.

**VRM with existing techniques.** We can also integrate the proposed vicinal level-1 labeling method into three representative models: the standard level-1 classifier (i.e., softmax classifier), a level-2 evidential model (EDL) [45], and the hyper-opinion-based H-EDL [44]. As shown in Table 4, our method can be seamlessly incorporated into existing techniques to enhance both OOD generalization and OOD detection. Several key observations are worth highlighting: First, vicinal level-1 labels are more compatible with level-2 distributional models than with point-estimate-based softmax classifiers. This is likely because softmax models lack the ability to express uncertainty over multiple plausible classes, whereas distributional models can better leverage the probabilistic nature of vicinal labels.

Moreover, our approach is particularly effective when combined with H-EDL [44], which explicitly captures the possibility of an instance belonging to multiple classes through feature-based hyper-opinions. The synergy between vicinal labels and hyper-opinion enables more accurate and continuous modeling of probability distributions.

Table 4: Level-1 and level-2 models with VRM.

| Method | Model generalization | | OOD detection |
|---|---|---|---|
| | ID-Acc | OOD-Acc | AUROC |
| MSP [18] | 95.06 | 74.75 | 89.83 |
| EDL [45] | 95.17 | 74.51 | 90.67 |
| H-EDL [44] | 95.04 | 73.84 | 92.27 |
| MSP w. Vic | $96.33_{+1.27}$ | $87.92_{+13.17}$ | $89.59_{-0.24}$ |
| EDL w. Vic | $96.18_{+1.01}$ | $88.73_{+14.22}$ | $93.08_{\mathbf{+2.39}}$ |
| H-EDL w. Vic | $96.43_{+1.39}$ | $88.63_{+14.79}$ | $93.89_{+1.72}$ |

**Ablation study.** As shown in Fig. 3, we investigate the impact of two hyperparameters: $\beta$ and $\beta_{\text{noise}}^{+}$. We begin by analyzing $\beta$. As $\beta$ increases, the corresponding Beta distribution becomes more peaked around 0.5, with narrower tails on both sides. This sharper concentration near 0.5

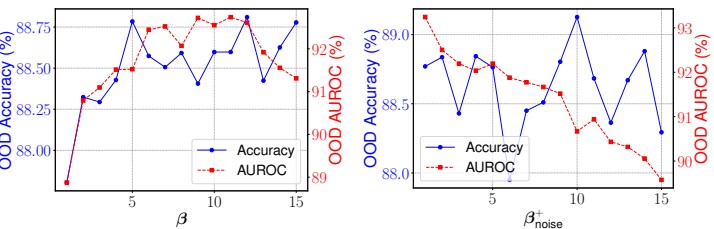

Figure 3: Ablation study on the hyperparameters.

facilitates improved OOD generalization, as supported by Theorem 2. However, an overly concentrated $\lambda$ distribution around 0.5 can hinder smooth sample mixing, which in turn may degrade OOD detection performance. Empirically, we find that setting $\beta$ to around 10 provides the best trade-off. For the analysis of $\beta_{\text{noise}}^{+}$, we first consider the case where $\beta_{\text{noise}}^{+} = \beta_{\text{noise}}^{-} = 1.0$. In this scenario, the resulting $\lambda$ values follow a uniform distribution over the interval [0, 1]. In this case, OOD detection performance reaches its peak, as smaller $\lambda$ values (according to Theorem 3) help suppress the rapid growth of Dirichlet concentration, thereby enhancing epistemic uncertainty estimation. However, as $\beta_{\text{noise}}^{+}$ increases, the sampled $\lambda$ values become increasingly concentrated near 1, which accelerates the Dirichlet concentration growth and compromises the model's ability to estimate epistemic uncertainty accurately. While our primary analysis focuses on their joint effect, we further isolate and analyze the contribution of each component through detailed ablation experiments in Appendix D.2.

## 6 Conclusion

This work addresses a key limitation of current EDL methods, namely their reliance on hard labels that ignore inherent label uncertainty. We propose a vicinal risk minimization framework that employs level-1 supervision through smoothed conditional label distributions. This approach enhances aleatoric uncertainty modeling and mitigates Dirichlet degeneration, also resulting in improved epistemic uncertainty estimation. Extensive experiments demonstrate consistent improvements across both out-of-distribution and selective classification benchmarks.

**Limitations.** While our method effectively alleviates the degeneration problem in evidential uncertainty estimation, it introduces two hyperparameters that control the distribution of the generated level-1 vicinal labels. The choice of these hyperparameters can influence the balance between aleatoric and epistemic uncertainty, similar to how the regularization strength in previous works based on entropy or Fisher information affects standard EDL methods. Although we empirically found our approach to be robust within a reasonable range of parameter values, developing adaptive or data-driven strategies to automatically calibrate the vicinal smoothing strength remains an important direction for future work.

## Acknowledgments and Disclosure of Funding

This work was supported by the National Natural Science Foundation of China (No. 62472315, 62476165). We thank Wei Liu and Xujing Zhou for their constructive discussions on this work.

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

# A   List of Symbols

A list of symbols used in the main paper as well as in the following supplementary material, most of symbols keep same as [2][3].

Table 5: Notation summary for the general, level-1, and level-2 learning settings.

| **General Symbols** | |
| --- | --- |
| $K$ | number of classes |
| $\mathcal{X}$ | instance space |
| $\mathcal{Y}$ | label space with hard labels $\{y_1, \ldots, y_K\}$ |
| $\mathcal{D}$ | training data $\{(\boldsymbol{x}^{(n)}, y^{(n)})\}_{n=1}^N \subset \mathcal{X} \times \mathcal{Y}$ |
| $P$ | data generating probability |
| $p(\cdot \mid \boldsymbol{x})$ | a conditional distribution on $\mathcal{Y}$, i.e., $p(y \mid \boldsymbol{x})$, represents the probability of observing $y$ given $\boldsymbol{x}$ |
| $\mathbb{P}(\mathcal{Y}), \mathbb{P}_1(\mathcal{Y})$ | the set of probability distributions on $\mathcal{Y}$ |
| $\Delta_K$ | the $K$-simplex, i.e., $\Delta_K := \{\boldsymbol{\theta} = (\theta_1, \ldots, \theta_K) \in [0,1]^K \mid \|\boldsymbol{\theta}\|_1 = 1\}$ |
| $\boldsymbol{\theta} = (\theta_1, \cdots, \theta_K)^\top$ | probability vector with $K$ singletons |
| **Level-1 Learning Setting** | |
| $\mathcal{H}_1$ | (level-1) hypothesis space consisting of hypothesis $h : \mathcal{X} \to \Delta_K$ |
| $L_1$ | loss function for level-1 hypothesis, i.e., $L_1 : \mathbb{P}_1(\mathcal{Y}) \times \mathcal{Y} \to \mathbb{R}$ |
| $R(\cdot)$ | risk or expected loss of a level-1 hypothesis (Eq.3) |
| $\hat{R}_{\mathrm{emp}}(\cdot)$ | empirical loss of a level-1 hypothesis (Eq. 4) |
| $\hat{h}$ | empirical risk minimiser, i.e., $\hat{h} = \arg\min_{h \in \mathcal{H}} \hat{R}_{\mathrm{emp}}(h)$ |
| $h^*$ | true risk minimiser or Bayes predictor, i.e., $h^* = \arg\min_{h \in \mathcal{H}} R(h)$ |
| **Level-2 Learning Setting** | |
| $\Delta_K^{(2)}$ | the set of distributions on simplex $\Delta_K$ |
| $\mathbb{P}_2(\mathcal{Y})$ | the set of distributions on $\mathbb{P}_1(\mathcal{Y})$ (the set of level-2 distributions) |
| $\mathcal{H}_2$ | (level-2) hypothesis, i.e., a mapping $h : \mathcal{X} \to \Delta_K^{(2)}$ |
| $Q$ | probability distribution on $\mathbb{P}_1(\mathcal{Y})$, i.e., an element of $\mathbb{P}_2(\mathcal{Y})$ |
| $L_2$ | loss function for level-2 hypothesis, e.g., $L_2 : \mathbb{P}_2(\mathcal{Y}) \times (\cdot) \to \mathbb{R}_+$ |
| $\hat{R}_{\mathrm{emp}}^{(2)}(\cdot)$ | empirical (level-2) loss of a level-2 hypothesis |
| $R^{(2)}(\cdot)$ | (level-2) risk or expected loss of a level-2 hypothesis |
| **Distributions** | |
| $\mathcal{N}(\mu, \sigma^2)$ | Gaussian distribuiton with location parameter $\mu$ and scale parameter $\sigma > 0$ |
| $\mathrm{Dir}(\boldsymbol{\alpha})$ | Dirichlet distribution with parameter $\boldsymbol{\alpha} \in \mathbb{R}_+^K$ |
| $\delta_y$ | Dirac measure at $y \in \mathcal{Y}$ (i.e. $\delta_y$ is an element of $\mathbb{P}_1(\mathcal{Y})$) |
| $\delta_p$ | Dirac measure at $p \in \mathbb{P}_1(\mathcal{Y})$ (i.e., $\delta_p$ is an element of $\mathbb{P}_2(\mathcal{Y})$) |
| **Entropy and Divergence** | |
| $H(\cdot)$ | Shannon Entropy of a categorical distribution |
| $\mathrm{KL}(\cdot, \cdot)$ | Kullback-Leibler divergence on $\mathbb{P}_2(\mathcal{Y}) \times \mathbb{P}_2(\mathcal{Y})$ |

# B   Proof of Theorem

**Theorem 1.** *For any level-1 loss function $L_1 : \mathbb{P}_1(\mathcal{Y}) \times \mathcal{Y} \to \mathbb{R}$ that satisfies $L_1(\mathbb{E}_{p \sim Dir(\boldsymbol{\alpha})} \boldsymbol{p}, \cdot) \leq \mathbb{E}_{p \sim Dir(\boldsymbol{\alpha})} L_1(\boldsymbol{p}, \cdot)$, (i.e., is a convex function), such as Brier score and the log-loss in Eq. 2, the empirical risk minimizer of a level-2 prediction is always a Dirac measure $\delta_p \in \mathbb{P}_2(\mathcal{Y})$ and the expectation of level-2 prediction is $\delta_y \in \mathbb{P}_1(\mathcal{Y})$. This result holds if the learner possesses a universal approximation property, allowing it to represent such a degenerate distribution.*

*Proof.* Let the empirical risk of a level-2 prediction $Q \in \mathbb{P}_2(\mathcal{Y})$ as

$$\hat{R}_{\text{emp}}^{(2)}(Q) = \frac{1}{N} \sum_{n=1}^{N} L_2\left(Q, y^{(n)}\right)$$

$$= \frac{1}{N} \sum_{n=1}^{N} \mathbb{E}_{\boldsymbol{p} \sim Q} L_1\left(\boldsymbol{p}, y^{(n)}\right). \tag{19}$$

By assumption on the level-1 loss $L_1$ (i.e. convexity), it holds that

$$\hat{R}_{\text{emp}}^{(2)}(Q) \geq \frac{1}{N} \sum_{n=1}^{N} L_1\left(\mathbb{E}_{\boldsymbol{p} \sim Q}[\boldsymbol{p}], y^{(n)}\right). \tag{20}$$

Let $\widetilde{Q}^{(N)}$ be the minimizer over all $Q \in \Delta_K^{(2)}$ of the right-hand side, then $\tilde{\boldsymbol{p}}^{(N)} = \mathbb{E}_{\boldsymbol{p} \sim \widetilde{Q}^{(N)}}[\boldsymbol{p}]$ is an element in $\Delta_K$. Define $\hat{Q}^{(N)} = \delta_{\tilde{\boldsymbol{p}}^{(N)}}$ and note that $\mathbb{E}_{\boldsymbol{p} \sim \hat{Q}^{(N)}}[\boldsymbol{p}] = \tilde{\boldsymbol{p}}^{(N)}$. Then,

$$\hat{R}_{\text{emp}}^{(2)}(\hat{Q}^{(N)}) = \frac{1}{N} \sum_{n=1}^{N} \mathbb{E}_{\boldsymbol{p} \sim \hat{Q}^{(N)}} L_1(\boldsymbol{p}, y^{(n)})$$

$$= \frac{1}{N} \sum_{n=1}^{N} L_1\left(\tilde{\boldsymbol{p}}^{(N)}, y^{(n)}\right) \tag{21}$$

$$= \frac{1}{N} \sum_{n=1}^{N} L_1\left(\mathbb{E}_{\boldsymbol{p} \sim \tilde{Q}^{(N)}}[\boldsymbol{p}], y^{(n)}\right).$$

This proves that the empirical level-2 risk is minimized by a Dirac distribution over a single level-1 prediction, i.e., $\hat{Q}^{(N)} = \delta_{\tilde{\boldsymbol{p}}^{(N)}}$, implying vanishing epistemic uncertainty. We now show that the corresponding level-1 prediction also collapses to a Dirac measure, indicating vanishing aleatoric uncertainty. Consider the empirical level-1 risk:

$$\hat{R}_{\text{emp}}^{(1)}(\boldsymbol{p}) = \frac{1}{N} \sum_{n=1}^{N} L_1(\boldsymbol{p}, y^{(n)}). \tag{22}$$

For any strictly proper loss function $L_1$ (e.g., Brier score, log-loss), it is uniquely minimized when $\boldsymbol{p} = \delta_{y^{(n)}}$, i.e., the one-hot encoding of the ground-truth label. That is,

$$\arg \min_{\boldsymbol{p} \in \Delta_K} L_1(\boldsymbol{p}, y^{(n)}) = \delta_{y^{(n)}}, \quad \text{with} \quad L_1\left(\delta_{y^{(n)}}, y^{(n)}\right) = 0. \tag{23}$$

Hence, the optimal level-1 predictor $\tilde{\boldsymbol{p}}^{(N)}$ that minimizes the empirical risk is

$$\tilde{\boldsymbol{p}}^{(N)} = \delta_{y^{(n)}}, \quad \text{for all } n. \tag{24}$$

It follows that the expected level-1 prediction under the optimal level-2 distribution is

$$\mathbb{E}_{\boldsymbol{p} \sim \hat{Q}^{(N)}} \boldsymbol{p} = \delta_{y^{(n)}}, \tag{25}$$

i.e., a one-hot distribution that assigns all probability mass to the ground-truth class. This indicates that aleatoric uncertainty also vanishes. □

Therefore, the empirical level-2 risk is minimized by a Dirac measure over a level-1 Dirac prediction

$$\hat{Q}^{(N)} = \delta_{\delta_{y^{(n)}}}. \tag{26}$$

This implies that:

- **Epistemic uncertainty vanishes**, since $Q$ is a Dirac measure.
- **Aleatoric uncertainty vanishes**, since the expected level-1 prediction under $Q$ is a one-hot vector.

This highlights a critical degeneracy of empirical risk minimization with strictly proper convex losses in the level-2 setting: it collapses all predictive uncertainty, providing no representation of uncertainty despite operating in a distribution-over-distributions framework.

**Proposition 1.** *Under the assumptions of Theorem 1, empirical risk minimization of level-2 prediction inevitably yields degenerate distributions $\delta_p \in \mathbb{P}_2(\mathcal{Y})$ and the expectation of the level-2 prediction is $\delta_y \in \mathbb{P}_1(\mathcal{Y})$. As a result, the model fails to provide any meaningful or disentangled representation of aleatoric or epistemic uncertainty.*

*Proof.* Assume that the optimal strategy under ERM is to collapse the Dirichlet distribution to a delta distribution centered on the one-hot vector $\delta_y$, i.e., $\text{Dir}(\boldsymbol{\alpha}) \to \delta_{\delta_y}$ as in Theorem 1. This degeneracy has consequences for uncertainty estimation. Consider the standard decomposition of predictive uncertainty in Dirichlet-based models as in Theorem 1, we have

$$\text{Total Uncertainty (TU)} \ = \ H\left[\mathbb{E}_{\boldsymbol{p} \sim \text{Dir}(\boldsymbol{\alpha})}\left[p(y \mid \boldsymbol{p})\right]\right], \tag{27}$$

$$\text{Aleatoric Uncertainty (AU)} \ = \ \mathbb{E}_{\boldsymbol{p}}\left[H\left[p(y \mid \boldsymbol{p})\right]\right], \tag{28}$$

$$\text{Epistemic Uncertainty (EU)} \ = \ \text{TU} - \text{AU}. \tag{29}$$

When the Dirichlet degenerates to $\delta_{\delta_y}$, both the expected predictive distribution and the samples from $\text{Dir}(\boldsymbol{\alpha})$ are deterministic, yielding

$$\text{TU} \to 0, \quad \text{AU} \to 0, \quad \text{EU} \to 0. \tag{30}$$

Thus, the model expresses neither AU nor EU, regardless of the true nature of the data distribution. Consequently, the level-2 model fails to provide any meaningful or disentangled representation of aleatoric or epistemic uncertainty. $\qquad\square$

**Theorem 2.** *Let the ground-truth level-1 label be denoted as $\boldsymbol{p}^*(\boldsymbol{x})$, and let the observed level-0 one-hot label $\delta_y(\boldsymbol{x})$ be a noisy realization of $\boldsymbol{p}^*(\boldsymbol{x})$ perturbed by input-dependent label noise $\boldsymbol{\mu}(\boldsymbol{x})$*

$$\delta_y(\boldsymbol{x}) = \boldsymbol{p}^*(\boldsymbol{x}) + \boldsymbol{\mu}(\boldsymbol{x}) \quad where \quad \boldsymbol{\mu}(\boldsymbol{x}) \sim \mathcal{N}(\mathbf{0}, \sigma^2 \boldsymbol{I}). \tag{31}$$

*Then, the test risk admits the following lower bound under mild regularity conditions*

$$R(\hat{h}; P) \geq C\sigma^2, \tag{32}$$

*where $C$ depends on the trace of the Hessian matrix of the loss function with respect to $\boldsymbol{p}$. Then, for the level-1 label with strong mixing, the bound can be tightened as*

$$R(\hat{h}; P) \geq C'\sigma^2, \tag{33}$$

*where $C'/C \approx \frac{1}{2\beta+1} + \frac{1}{2} < 1 \ (\forall \beta \gg 1/2)$, indicating a reduced sensitivity of the test risk to input-dependent noise.*

*Proof.* We suppose the label noise $\boldsymbol{\mu}$ follows an isotropic Gaussian distribution as $\mathcal{I}$-EDL [12]:

$$\boldsymbol{\mu} \sim \mathcal{N}(0, \sigma^2 \boldsymbol{I}). \tag{34}$$

Then, even if the optimization loss $R(\hat{h}; \mathcal{D})$ is minimized (or approaches zero), the population loss $R(\hat{h}; \mathcal{P})$ will have an irreducible component that is at least on the order of $\sigma^2$. As we assume that the training labels $y$ are generated from the true labels $\boldsymbol{p}^*$ with added noise:

$$\delta_y(\boldsymbol{x}) = \boldsymbol{p}^*(\boldsymbol{x}) + \boldsymbol{\mu}(\boldsymbol{x}), \tag{35}$$

where $\boldsymbol{\mu}(\boldsymbol{x}) \sim \mathcal{N}(0, \sigma^2 \boldsymbol{I})$. The expected test loss can be expressed as

$$R(\hat{h}; P) := \mathbb{E}_{(\boldsymbol{x}, y) \sim P}\left[L_2\big(\hat{h}(\boldsymbol{x}), y\big)\right]. \tag{36}$$

Since the label itself is affected by noise, we can decompose the expectation as

$$\mathbb{E}\left[L_2\big(\hat{h}(\boldsymbol{x}), \delta_y\big)\right] = \mathbb{E}\left[L_2\big(\hat{h}(\boldsymbol{x}), \boldsymbol{p}^* + \boldsymbol{\mu}\big)\right]. \tag{37}$$

Using a second-order Taylor expansion to approximate the loss function:

$$\ell\big(\hat{h}(\boldsymbol{x}), \boldsymbol{p}^* + \boldsymbol{\mu}\big) \approx L_2\big(\hat{h}(\boldsymbol{x}), \boldsymbol{p}^*\big) + \langle \nabla L_2, \boldsymbol{\mu}\rangle + \frac{1}{2}\boldsymbol{\mu}^\top H \boldsymbol{\mu}. \tag{38}$$

where $\boldsymbol{H}$ represents the Hessian matrix of the loss function $L_2(\hat{h}(\boldsymbol{x}), \boldsymbol{p}^*)$ w.r.t. $\boldsymbol{p}^*$, defined as

$$\boldsymbol{H} = \nabla^2 L_2(\hat{h}(\boldsymbol{x}), \boldsymbol{p}^*), \tag{39}$$

and $\langle \nabla L_2, \boldsymbol{\mu} \rangle$ is the inner product between the gradient of the loss function and the noise vector $\boldsymbol{\mu}$:

$$\langle \nabla L_2, \boldsymbol{\mu} \rangle = \sum_k^K \frac{\partial L_2}{\partial_k} \mu_k. \tag{40}$$

Since the noise $\boldsymbol{\mu}$ follows a zero-mean Gaussian distribution, the expectation of the first-order term vanishes:

$$\mathbb{E}[\langle \nabla L_2, \boldsymbol{\mu} \rangle] = 0, \tag{41}$$

while the expectation of the second-order term is given by the noise covariance:

$$\mathbb{E}[\boldsymbol{\mu}^\top \boldsymbol{H} \boldsymbol{\mu}] = \sigma^2 \operatorname{Tr}(\boldsymbol{H}). \tag{42}$$

Thus, the lower bound of the test loss can be approximated as

$$R(\hat{h}; \mathcal{P}) \geq C\sigma^2, \tag{43}$$

where $C$ depends on the trace of the Hessian matrix. We then show that incorporating VRM leads to a lower test risk. Let the original label noise $\boldsymbol{\mu}^{(n)}, \boldsymbol{\mu}^{(m)} \sim \mathcal{N}(0, \sigma^2 \boldsymbol{I})$ be i.i.d. After vicinal interpolation, the noise in vicinal labels becomes

$$\tilde{\boldsymbol{\mu}} = \lambda \boldsymbol{\mu}^{(n)} + (1 - \lambda) \boldsymbol{\mu}^{(m)}, \tag{44}$$

with variance

$$\mathbb{E} \|\tilde{\boldsymbol{\mu}}\|^2 = \lambda^2 \sigma^2 + (1 - \lambda)^2 \sigma^2 = \sigma^2 \left[ \lambda^2 + (1 - \lambda)^2 \right]. \tag{45}$$

When $\lambda \sim \operatorname{Beta}(\beta, \beta)$, the expected variance is

$$\mathbb{E}_\lambda \left[ \lambda^2 + (1 - \lambda)^2 \right] = 2\mathbb{E}[\lambda^2] - 2\mathbb{E}[\lambda] + 1. \tag{46}$$

Using properties of Beta distribution $\mathbb{E}[\lambda] = \frac{1}{2}$ and $\operatorname{Var}(\lambda) = \frac{1}{4(2\beta+1)}$, we obtain

$$\mathbb{E}[\lambda^2] = \operatorname{Var}(\lambda) + (\mathbb{E}[\lambda])^2 = \frac{1}{4(2\beta + 1)} + \frac{1}{4}. \tag{47}$$

Substituting yields

$$\mathbb{E}_\lambda \left[ \lambda^2 + (1 - \lambda)^2 \right] = \frac{1}{2\beta + 1} + \frac{1}{2} < 1 \quad (\forall \beta \gg 1/2). \tag{48}$$

Thus, the effective noise variance after Mixup is $k\sigma^2$, where $k = \frac{1}{2\beta+1} + \frac{1}{2} < 1$, significantly lower than the original $\sigma^2$. Substituting into the theorem's lower bound gives

$$R(\hat{h}; P) \geq C \cdot k\sigma^2 < C\sigma^2. \tag{49}$$

Although distribution of the noise $\boldsymbol{\mu}$ is unknown; and assumptions about it are modeling questions, most statistical methods rely on certain mathematical conditions, known as regularity assumptions, to ensure their validity. In our proof, i.e., we assume that $\boldsymbol{\mu}$ follows an additive Gaussian noise. $\qquad \square$

**Theorem 3.** *Let $\lambda$ be the mixing hyperparameter defined in Eq. 13. Consider the optimization of the Dirichlet parameters $\boldsymbol{\alpha}$ in Eq. 14. For samples where $\alpha_k \leq \alpha_j$ ($\forall j \neq k$) with lower belief assigned to the ground-truth $k$ class, the following properties hold*

- *The update to the Dirichlet concentration for the ground-truth class $\Delta\alpha_k$, increases monotonically with $\lambda$.*

- *The updates to the Dirichlet concentrations for the non-ground-truth classes $\Delta\alpha_{j \neq k}$, decrease monotonically with $\lambda$.*

- *The total increase in Dirichlet concentration, denoted $\Delta S$, increases monotonically with $\lambda$.*

*Proof.* Lets take the $L_1$ loss as cross-entropy loss for example, which calculate loss between the sampled $\boldsymbol{p}$ from $\text{Dir}(\boldsymbol{\alpha})$ with $\tilde{y}$. Then, we derive the following analytical form of $\mathcal{L}_{\text{edl}}$ as

$$
\mathcal{L}_{\text{edl}}(\boldsymbol{\alpha}, \tilde{\boldsymbol{y}}) = \int \left[ \sum_{j=1}^{K} -\tilde{y}_j \log\left(p_j\right) \right] \frac{1}{\mathrm{B}\left(\boldsymbol{\alpha}\right)} \prod_{j=1}^{K} p_j^{\alpha_j - 1} d\boldsymbol{p}
$$

$$
= \sum_{j=1}^{K} \tilde{y}_j \left( \psi\left(S\right) - \psi\left(\alpha_j\right) \right) \tag{50}
$$

where $S = \sum_{j=1}^{K} \alpha_j$. Then, with gradient descent, the update of $\alpha_j$, we denote as $-\eta \frac{\partial \mathcal{L}_{\text{edl}}}{\alpha_j}$, where $\eta$ is the learning rate. Let $j$ denote the index of class, we have

$$
\frac{\partial \mathcal{L}_{\text{edl}}(\boldsymbol{\alpha}, \tilde{\boldsymbol{y}})}{\alpha_j} = \psi_1(S) \cdot \sum_{i=1}^{K} \tilde{y}_i - \tilde{y}_j \psi_1(\alpha_j) = \psi_1(S) - \tilde{y}_j \psi_1(\alpha_j) \qquad \text{as} \qquad \sum_{i=1}^{K} \tilde{y}_i = 1 \tag{51}
$$

where $\psi_1$ is the trigamma function, which is a positive, monotonic decreasing function. Then, we have the updates of $\alpha_j$ as Eq. 52 with the negative gradient descent update

$$
\Delta \alpha_j = -\eta \left[ \psi_1(S) - \tilde{y}_j \psi_1(\alpha_j) \right] \tag{52}
$$

As the vicinal label is obtained by $\tilde{\boldsymbol{y}} = \lambda \boldsymbol{y}^{(n)} + (1 - \lambda) \cdot \left[ \frac{1}{K}, \ldots, \frac{1}{K} \right]$, we can also express the smoothed target labels explicitly as

$$
\tilde{y}_k = \lambda + \frac{1 - \lambda}{K}, \quad \tilde{y}_j = \frac{1 - \lambda}{K}, \tag{53}
$$

where $k$ denotes the index of ground-truth class. By substituting Eq. 53 into Eq. 52, we have

$$
\Delta \alpha_k = -\eta \left[ \psi_1(S) - \left( \lambda + \frac{1 - \lambda}{K} \right) \psi_1(\alpha_k) \right] \tag{54}
$$

$$
\Delta \alpha_j = -\eta \left[ \psi_1(S) - \frac{1 - \lambda}{K} \psi_1(\alpha_j) \right], \qquad j \neq k \tag{55}
$$

and

$$
\Delta S = \sum_{j=1}^{K} \Delta \alpha_j = -\eta \left[ K \psi_1(S) - \left( \lambda + \frac{1 - \lambda}{K} \right) \psi_1(\alpha_k) - \frac{1 - \lambda}{K} \sum_{j \neq k} \psi_1(\alpha_j) \right]. \tag{56}
$$

To analyze how $\lambda$ affects $\Delta \alpha_k$, $\Delta \alpha_j$, and $\Delta S$, consider the derivatives as follows.

$$
\frac{\partial \Delta \alpha_k}{\partial \lambda} = \eta \left( 1 - \frac{1}{K} \right) \psi_1(\alpha_k) > 0 \tag{57}
$$

and

$$
\frac{\partial \Delta \alpha_j}{\partial \lambda} = -\eta \frac{1}{K} \psi_1(\alpha_j) < 0, \qquad j \neq k \tag{58}
$$

and

$$
\frac{\partial \Delta S}{\partial \lambda} = \sum_{j=1}^{K} \frac{\partial \Delta \alpha_j}{\partial \lambda}
$$

$$
= \eta \left[ \left( 1 - \frac{1}{K} \right) \psi_1(\alpha_k) - \frac{1}{K} \sum_{j \neq k} \psi_1(\alpha_j) \right]
$$

$$
= \eta \sum_{t=0}^{T} \left[ \psi_1(\alpha_k) - \frac{1}{K} \sum_{j=1}^{K} \psi_1(\alpha_j) \right]
$$

$$
= \frac{\eta}{K} \sum_{j=1}^{K} \left( \psi_1(\alpha_k) - \psi_1(\alpha_j) \right) \tag{59}
$$

As the label smooth process takes the following

$$\tilde{x} = \lambda \boldsymbol{x}^{(n)} + (1 - \lambda)\boldsymbol{x}^{(m)}, \qquad \tilde{y} = \lambda y^{(n)} + (1 - \lambda)\left[\frac{1}{K}, ..., \frac{1}{K}\right]. \tag{60}$$

This analysis reveals how label smoothing influences the accumulation of Dirichlet strength. When the model is not yet confident in the true class $k$, its corresponding Dirichlet strength $\alpha_k$ is relatively small. Given that the trigamma function $\psi_1(x)$ is monotonically decreasing, a smaller $\alpha_k$ results in $\psi_1(\alpha_k)$ being larger than the average trigamma value across all classes (i.e., $\psi_1(\alpha_k) > \frac{1}{K}\sum_{j=1}^{K}\psi_1(\alpha_j)$). Consequently, the derivative $\frac{\partial \Delta S}{\partial \lambda}$ becomes positive. This positive derivative indicates that a decrease in $\lambda$ (which corresponds to an increased degree of label smoothing) will lead to a smaller increment $\Delta S$, thus slowing the growth of the total Dirichlet strength $S$.

$\square$

## C  Uncertainty Measures

### C.1  Uncertainty Decomposition in Dirichlet-Based Models

A fundamental identity in information theory is that the Shannon entropy of a random variable $X$ can be additively decomposed into the mutual information between $X$ and $Y$, and the conditional entropy of $X$ given $Y$ [1]:

$$H(X) = I(X; Y) + H(X \mid Y) \tag{61}$$

Follow this idea, Prior Networks [36] propose a method to explicitly model and decompose predictive uncertainty into two components: *aleatoric uncertainty* and *epistemic uncertainty*. This is achieved by treating the output of the classifier as the parameters of a Dirichlet distribution over categorical class distributions. Given a Dirichlet distribution parameterized by $\boldsymbol{\alpha} = (\alpha_1, \ldots, \alpha_K)$ over the probability simplex $\Delta_K$, the expected predictive distribution over class labels is:

$$p(y = j \mid \boldsymbol{x}) = \mathbb{E}_{\boldsymbol{p} \sim \text{Dir}(\boldsymbol{\alpha})}[p_j] = \frac{\alpha_j}{S}, \quad \text{where } S = \sum_{j=1}^{K} \alpha_j \tag{62}$$

The total uncertainty in the prediction is measured by the Shannon entropy of the expected categorical distribution conditioned :

$$H_{\text{total}}[p(y|\boldsymbol{p})] = \mathbb{E}_{\boldsymbol{p} \sim \text{Dir}(\boldsymbol{\alpha})}[p(y \mid \boldsymbol{p})] = -\sum_{j=1}^{K} \frac{\alpha_j}{S} \log \frac{\alpha_j}{S}, \tag{63}$$

### C.2  Conditional Entropy

Aleatoric uncertainty corresponds to the expected entropy of the categorical distributions sampled from the Dirichlet prior, commonly referred to as the *conditional entropy*

$$\mathbb{E}_{\boldsymbol{p} \sim \text{Dir}(\boldsymbol{\alpha})}\left[H[p(y \mid \boldsymbol{p})]\right] = \mathbb{E}_{\boldsymbol{p}}\left[-\sum_{j=1}^{K} p_j \log p_j\right]$$

$$= -\sum_{j=1}^{K} \frac{\alpha_j}{S}\left(\psi(\alpha_j + 1) - \psi(S + 1)\right) \tag{64}$$

$$= \psi(S + 1) - \sum_{j=1}^{K} \frac{\alpha_j}{S}\psi(\alpha_j + 1)$$

### C.3  Mutual Information

Epistemic uncertainty can be measured by the *mutual information* between predictions and the Dirichlet parameters, capturing uncertainty about the model itself:

$$\text{MI}(y, \boldsymbol{p}) = H_{\text{total}}[p(y|\boldsymbol{p})] - \mathbb{E}_{\boldsymbol{p} \sim \text{Dir}(\boldsymbol{\alpha})}[H[p(y \mid \boldsymbol{p})]]. \tag{65}$$

This mutual information quantifies how much of the total uncertainty arises from uncertainty in the model parameters (i.e., distribution over categorical distributions), and thus reflects *epistemic uncertainty*.

$$\underbrace{\mathrm{MI}[\boldsymbol{y}, \boldsymbol{p}]}_{\text{Epistemic Uncertainty}} \approx \underbrace{H\left[\mathbb{E}_{\boldsymbol{p} \sim \mathrm{Dir}(\boldsymbol{\alpha})}[p(y|\boldsymbol{p})]\right]}_{\text{Total Uncertainty}} - \underbrace{\mathbb{E}_{\boldsymbol{p} \sim \mathrm{Dir}(\boldsymbol{\alpha})}[H[p(y|\boldsymbol{p})]]}_{\text{Aleatoric Uncertainty}}$$

$$= -\sum_{j=1}^{K} \frac{\alpha_j}{S} \ln \frac{\alpha_j}{S} + \sum_{j=1}^{K} \frac{\alpha_j}{S} \left(\psi(\alpha_j + 1) - \psi(S + 1)\right) \quad (66)$$

$$= -\sum_{j=1}^{K} \frac{\alpha_j}{S} \left(\ln \frac{\alpha_j}{S} - \psi(\alpha_j + 1) + \psi(S + 1)\right).$$

### C.4 Differential Entropy

The *differential entropy* is defined as

$$\mathrm{ENT}(\mathrm{Dir}(\boldsymbol{p} \mid \boldsymbol{\alpha})) = -\int_{\Delta_K} \mathrm{Dir}(\boldsymbol{p} \mid \boldsymbol{\alpha}) \log \mathrm{Dir}(\boldsymbol{p} \mid \boldsymbol{\alpha}) \, d\boldsymbol{p}, \quad (67)$$

where $\Delta_K$ denotes the probability simplex. The closed-form expression is given by

$$\mathrm{ENT}(\mathrm{Dir}(\boldsymbol{p} \mid \boldsymbol{\alpha})) = \log B(\boldsymbol{\alpha}) + (S - K)\psi(S) - \sum_{j=1}^{K}(\alpha_j - 1)\psi(\alpha_j), \quad (68)$$

Differential entropy is also a prevalent measure of epistemic uncertainty, where a lower value indicates that the model yields a sharper distribution, and a higher value means a more uniform Dirichlet distribution.

### C.5 Vacuity of Evidence

For EDL [45], RED [40], $\mathcal{I}-$EDL [12], R-EDL [9], H-EDL [44], which grounded in Subjective Logic [21] and DS-Theory [11]. Subjective Logic provides a principled framework for modeling predictive uncertainty by interpreting the output of a neural network as an *opinion*—a structured representation of uncertainty over a discrete set of classes. Unlike conventional classifiers that output categorical probabilities, EDL models produce non-negative evidence values $\boldsymbol{e} = [e_1, e_2, \ldots, e_K]$ for each of the $K$ classes. These evidence values parameterize a Dirichlet distribution $\mathrm{Dir}(\boldsymbol{\alpha})$, where $\alpha_j = e_j + 1$. In Subjective Logic, an opinion over a finite domain is characterized by three components: the belief mass $b_j$, the base rate $a_j$, and the uncertainty mass $u$, satisfying:

$$b_j + u \cdot a_j = \mathbb{E}[p_j], \quad \text{and} \quad \sum_{j=1}^{K} b_j + u = 1 \quad (69)$$

where $p_j$ denotes the probability assigned to class $j$. These quantities relate to the Dirichlet parameters as follows: The belief mass $b_k$ is proportional to the evidence for class $k$:

$$b_k = \frac{e_k}{S}, \quad \text{where } S = \sum_{j=1}^{K}(e_j + 1) = \sum_{j=1}^{K} \alpha_j \quad (70)$$

The base rate $a_k$ is typically assumed to be uniform, i.e., $a_k = 1/K$. The uncertainty mass $u$ is defined as:

$$u = \frac{K}{\sum_{j=1}^{K} \alpha_j} = \frac{K}{S} \quad (71)$$

This uncertainty mass $u$ is referred to as vacuity in EDL literature, and it quantifies the degree of epistemic uncertainty due to a lack of evidence. When the total evidence is low (e.g., under out-of-distribution or ambiguous inputs), $S$ becomes small and vacuity $u$ approaches 1, indicating that the model abstains from committing belief to any specific class. Conversely, high total evidence yields a low vacuity, reflecting confident predictions based on strong feature-based support. This opinion-based interpretation highlights the epistemic nature of uncertainty in EDL and differentiates it from aleatoric uncertainty captured by distributional spread in conventional probabilistic models.

# D  Additional Experimental Details

## D.1  Implementation Details

Since different baseline methods involve distinct activation functions and regularization terms, we provide detailed implementation settings below.

**EDLs based on Subjective Logic.** For EDL [45], we adopt the mean squared error (MSE) loss, also known as the *barrier score*. For $\mathcal{I}$-EDL [12], we follow their original paper and use the Fisher-MSE loss, setting the Fisher information regularization weight to 0.05. The activation function is Softplus, as specified in their implementation. For R-EDL [9], we follow the settings in the original paper and set the prior strength to 0.8 for the CIFAR datasets and using the MSE loss variant without the variance minimization term. For all three methods (EDL, $\mathcal{I}$-EDL, and R-EDL), the KL divergence term which aims to remove misleading evidence with an annealing weight schedule of $\lambda_t = \min(\text{epoch\_idx}/10, 1)$ . The KL divergence term which is used to regularize the predicted Dirichlet distribution by encouraging it to stay close to a non-informative prior for incorrect classes, typically $\text{Dir}(\boldsymbol{p} \mid \boldsymbol{1})$, where each class has a concentration parameter of 1. The KL divergence between the predicted Dirichlet distribution $\text{Dir}(\boldsymbol{p} \mid \bar{\boldsymbol{\alpha}})$ and the uniform Dirichlet prior

$$
\begin{aligned}
\mathcal{L}_{\text{KL}} &= \text{KL}[\text{Dir}(\boldsymbol{p} \mid \bar{\alpha}) \| \text{Dir}(\boldsymbol{p} \mid \boldsymbol{1})] \\
&= \log\left( \frac{\Gamma\left(\sum_{j=1}^{K} \alpha_j\right)}{\prod_{j=1}^{K} \Gamma(\alpha_j)} \right) + \sum_{j=1}^{K} (\alpha_j - 1) \left[ \psi(\alpha_j) - \psi\left( \sum_{j=1}^{K} \alpha_j \right) \right]
\end{aligned}
\tag{72}
$$

**PriorNets.** For KL-PN [36] and RKL-PN [37], we set the target class Dirichlet concentration parameter $\alpha_k$ to 200. Since both methods require out-of-distribution (OOD) samples during training to constrain their predicted Dirichlet distributions, we follow the setup in [12, 9] and use random noise as the OOD dataset to ensure a fair comparison.

**Our method.** For our method, the non-negative activation function $\sigma$ is set to Softplus for CIFAR-10. For CIFAR-100, due to the large zero-evidence regions observed in prior work [40], we warm up the model using an Exponential activation for the first 10 epochs to help the model avoid these regions, and then switch to a Softplus activation for the remainder of training.

## D.2  Further Ablation Study on Vicinal Supervision and Noise Augmentation

To clarify the individual contributions of the two components in the total loss (Eq. 15), we conduct an ablation study isolating the effects of *vicinal supervision* ($\mathcal{L}_{\text{vicinal}}$) and *noise augmentation* ($\mathcal{L}_{\text{noise}}$).

**Effect of Vicinal Supervision.** We first isolate the influence of $\beta$ by removing the noise augmentation term $\mathcal{L}_{\text{noise}}$. Table 6 summarizes the results as $\beta$ varies.

Table 6: Ablation on vicinal supervision by varying $\beta$ while removing $\mathcal{L}_{\text{noise}}$.

| $\beta$ | $\beta_{\text{noise}}^{+}$ | E-AURC ↓ | OOD AUROC ↑ | OOD Acc ↑ | ID Acc ↑ |
|---|---|---|---|---|---|
| – | – | 56.54±3.98 | 90.67±0.35 | 74.51±0.49 | 95.17±0.18 |
| 0.2 | – | 58.10±6.04 | 89.77±0.63 | 76.39±0.55 | 95.31±0.08 |
| 0.4 | – | 49.44±4.40 | 90.28±0.05 | 77.82±0.61 | 95.61±0.16 |
| 1.0 | – | 45.18±2.19 | 91.11±0.40 | 79.13±0.32 | 96.01±0.02 |
| 5.0 | – | 42.94±4.37 | 91.68±0.48 | 79.75±1.02 | 95.89±0.21 |
| 10.0 | – | 39.15±4.33 | 91.73±0.14 | 80.41±1.23 | 95.86±0.16 |

We observe three consistent trends: (1) **Improved aleatoric uncertainty estimation.** Increasing $\beta$ yields a monotonic decrease in E-AURC, from 56.54 to 39.15, indicating better calibration for selective classification. (2) **Enhanced OOD detection.** Despite being designed for aleatoric calibration, vicinal supervision also improves OOD AUROC, suggesting stronger discrimination between ID and OOD samples due to its regularizing effect on the data manifold. (3) **Improved generalization.** Both OOD and ID accuracies increase with $\beta$, supporting Theorem 2 that stronger mixup enhances generalization across both seen and unseen distributions.

**Effect of Noise Augmentation.** Next, we isolate $\mathcal{L}_{\text{noise}}$ by setting $\mathcal{L} = \mathcal{L}_{\text{noise}}$ and varying $\beta_{\text{noise}}^{+}$, while fixing $\beta_{\text{noise}}^{-} = 1.0$. The results are shown in Table 7.

Table 7: Ablation on noise augmentation by varying $\beta_{\text{noise}}^{+}$ while removing $\mathcal{L}_{\text{vicinal}}$.

| $\beta$ | $\beta_{\text{noise}}^{+}$ | OOD AUROC $\uparrow$ | E-AURC $\downarrow$ | OOD Acc $\uparrow$ | ID Acc $\uparrow$ |
|---|---|---|---|---|---|
| – | – | 90.67±0.35 | 56.54±3.98 | 74.51±0.49 | 95.17±0.18 |
| – | 0.2 | 90.95±0.32 | 56.11±4.92 | 74.32±0.43 | 95.00±0.02 |
| – | 0.4 | 91.18±0.42 | 62.81±10.15 | 73.76±0.67 | 95.03±0.21 |
| – | 1.0 | 91.85±0.24 | 57.22±6.14 | 73.96±0.43 | 95.17±0.09 |
| – | 5.0 | 90.74±0.20 | 55.95±6.40 | 74.67±0.75 | 95.02±0.20 |
| – | 10.0 | 90.37±0.55 | 60.17±2.85 | 73.42±0.53 | 94.86±0.41 |
| 10.0 | 1.0 | **93.08±0.33** | **29.40±2.16** | **88.73±0.13** | **96.18±0.13** |

From Table 7, we draw three conclusions: (1) **Limited effect of isolated $\beta_{\text{noise}}^{+}$.** Varying $\beta_{\text{noise}}^{+}$ alone causes only minor fluctuations in E-AURC and accuracy, suggesting that noise augmentation without vicinal supervision is insufficient for consistent gains. (2) **Importance of balanced noise intensity.** Too small $\beta_{\text{noise}}^{+}$ leads to excessive perturbations that harm learning, while too large values make the sampled $\lambda$ concentrate near 1, effectively disabling noise augmentation. $\beta_{\text{noise}}^{+} = 1.0$ yields the best balance. (3) **Synergistic effect.** The best overall performance is achieved when both components are combined ($\beta = 10.0$, $\beta_{\text{noise}}^{+} = 1.0$), achieving the highest OOD AUROC (93.08%), lowest E-AURC (29.40), and best ID/OOD accuracies, demonstrating the complementary benefits of vicinal supervision and noise augmentation.

# E  Discussions

## E.1  Why do some baseline methods perform poorly on CIFAR-100?

For models like EDLs [45, 9] and PriorNets [36, 37] that require Dirichlet concentrations of incorrect classes to approach zero, we observe that they struggle to converge when the number of classes is large (e.g., $K = 100$). Since the original papers do not provide CIFAR-100 experimental settings, we adopt the same configurations as used for CIFAR-10, which may limit their performance.

## E.2  Social Impact

Our work addresses the challenges of uncertainty estimation, out-of-distribution (OOD) detection, and OOD generalization, which are critical for ensuring the safety, reliability, and fairness of machine learning systems in real-world applications. By improving models' ability to recognize and appropriately respond to unfamiliar or ambiguous inputs, our methods help reduce the risk of overconfident mispredictions in high-stakes domains such as healthcare, autonomous driving, and finance. These advances have the potential to increase trust in AI systems and support more responsible deployment practices. Moreover, enhanced OOD generalization may help mitigate performance disparities when models are applied across diverse populations and settings.

