# OpenReview forum: "Vicinal Label Supervision for Reliable Aleatoric and Epistemic Uncertainty Estimation"
_NeurIPS.cc/2025/Conference — NeurIPS 2025 poster_

### Official Review · Reviewer_XHmk · 2025-06-23

**Clarity:** 3
**Significance:** 3
**Originality:** 2
**Rating:** 4
**Confidence:** 5

**Summary:**

This paper adresses a limitation in a class of deep learning uncertainty quantification (UQ) methods referred to as Evidential Deep Learning (EDL) methods, which rely on level-0 hard labels for supervision. The authors argue that hard labels do not reflect inherent data noise (aleatoric uncertainty). In practice, they observe that this mismatch encourages EDL models to produce degenerate Dirichlet distributions that collapse into near-deterministic outputs, leading to poor uncertainty quantification.

**Questions:**

A concern with this paper is that the experiments are not sufficiently convincing, as the authors seem eager to claim that their method establishes new standards in OOD input detection. Generally speaking, UQ methods are not explicitly designed for outlier/OOD input detection but may develop this capability as a byproduct. Specifically, the benchmark should include at least one UQ method known to exhibit strong outlier-detection performance, such as Liu et al. (2020). However, the experiments do validate the added value of their method compared to EDL for this use case.

(Liu et al. 2020) Liu, J., Lin, Z., Padhy, S., Tran, D., Bedrax Weiss, T., & Lakshminarayanan, B. (2020). Simple and principled uncertainty estimation with deterministic deep learning via distance awareness. Advances in neural information processing systems, 33, 7498-7512.

Another important aspect to be addressed in the experiments is evaluating other use cases for quantified uncertainties, particularly whether those uncertainties align with classification errors. Indeed, one generally expect higher levels of uncertainties in the regions of the input space where the model fails to predict the correct class labels.

Besides, many competing methods in the benchmark exhibit terrible performances. Could the authors comment on why they fail to so dramatically ?

The main contribution of the proposed modification to EDL is its ability to capture the input distribution (as Fig. 1 suggests) and leverage it to assign high uncertainty in low-probability regions of the input space. Yet this emerging capability is neither fully explored nor theoretically explained. Based on the authors’ comments that they use a large value for β, I suspect that most mixed samples lie near class boundaries; thus, it is unclear why the model would also learn to assign high uncertainty to inputs far from the training samples.

Minor: In Figure 1, unless I’m mistaken, the displayed uncertainty-scoring function is not specified.

**Ethical Concerns:**

["NO or VERY MINOR ethics concerns only"]

**Final Justification:**

I thank the authors for addressing the raised concerns. In light of the additional empirical and critical evidence provided by the authors, I raised my score by one unit.

**Limitations:**

The limitations are addressed very briefly (one sentence - 2 lines) in the conclusion. Usually, one would expect at least a whole paragraph.

**Quality:**

2

**Strengths And Weaknesses:**

Pros :
- Avoiding EDL degenerate solutions seems to be achieved by the proposed method
- The method is simple in its construction and thus does not add significant complexity on top of EDL algorithms.
- The efficiency of the method is theoretically grounded

Cons :
- The empirical comparisons in the experimental section omit important state-of-the-art works leveraging UQ methods to detect outliers/OOD inputs.
- The experiments focus on OOD detection uses cases and do not examine whether quantified uncertainties align with incorrect predictions, as is commonly done
- [Minor] The experiments involve training runs on two vision datasets and one toy dataset. The paper would gain in impact with a more solid evaluation involving more training datasets and other data modalities.

---

> ### Author Rebuttal · Authors · 2025-07-30
>
> # Response to Weakness
> ## Reponse to Comment 1
> Thank you for your valuable comment. We extend our experiments to include four SOTA UQ-based OOD detection methods—DUQ [1], DDU [2], DUE [3], and SNGP [4]. Our method achieves competitive performance, especially on CIFAR-100, without relying on additional techniques like spectral normalization adopted by competitive methods DDU and SNGP. This highlights the effectiveness of using level-1 vicinal labels to improve uncertainty estimation based OOD detection.
>
> #### → ID: CIFAR-10
>
> | Method | → CIFAR-100 | → Tiny | → MNIST | → SVHN | → Textures | → Places365 | CIFAR-10 Acc |
> |--------|-------------|--------|---------|--------|------------|-------------|--------------|
> | DUQ [1]  | 84.60 $\pm$   1.04 | 86.16 $\pm$  0.92 | 92.34 $\pm$  1.25 | 91.36 $\pm$ 1.26 | 86.57 $\pm$ 1.27 | 84.26 $\pm$ 0.93 | 93.60 $\pm$ 0.39 |
> | DDU [2]  | **89.23 $\pm$ 0.33** | 91.28 $\pm$ 0.28 | 95.69 $\pm$ 0.99 | 93.53 $\pm$ 1.43 | 92.64 $\pm$ 0.31 | 91.55 $\pm$ 0.34 | 95.35 $\pm$ 0.05 |
> | DUE [3]  | 86.00 $\pm$ 0.78 | 88.40 $\pm$ 0.76 | 92.34 $\pm$ 1.48 | 91.70 $\pm$ 1.63 | 89.25 $\pm$ 1.16 | 88.89 $\pm$ 0.77 | 94.98 $\pm$ 0.15 |
> | SNGP [4] | 88.14 $\pm$ 1.18 | 90.38 $\pm$ 1.00 | 93.18 $\pm$ 1.20 | 92.16 $\pm$ 2.40 | **92.88 $\pm$ 1.37** | 91.21 $\pm$ 1.20 | 94.63 $\pm$ 0.18 |
> | **Ours** | 89.09 $\pm$ 0.19 | **91.81 $\pm$ 0.22** | **97.32 $\pm$ 0.42** | **96.20 $\pm$ 0.32** | 92.51 $\pm$ 0.73 | **91.61 $\pm$ 0.15** | **96.18 $\pm$ 0.13** |
>
> #### → ID: CIFAR-100
>
> | Method | → CIFAR-10 | → Tiny | → MNIST | → SVHN | → Textures | → Places365 | CIFAR-100 Acc |
> |--------|------------|--------|---------|--------|------------|-------------|----------------|
> | DUQ [1]  | 51.20 $\pm$ 1.84 | 53.60 $\pm$ 2.67 | 39.44 $\pm$ 13.20 | 61.47 $\pm$ 7.64 | 57.73 $\pm$ 5.66 | 50.16 $\pm$ 3.13 | 1.66 $\pm$ 0.39 |
> | DDU [2]  | 68.14 $\pm$ 1.63 | 78.64 $\pm$ 1.57 | 79.69 $\pm$ 6.56 | 76.02 $\pm$ 3.98 | **83.00 $\pm$ 1.01** | 74.53 $\pm$ 1.70 | 78.05 $\pm$ 0.96 |
> | DUE [3]  | 50.30 $\pm$ 1.54 | 49.97 $\pm$ 1.39 | 49.91 $\pm$ 1.43 | 49.91 $\pm$ 1.02 | 50.02 $\pm$ 1.65 | 50.12 $\pm$ 1.01 | 1.06 $\pm$ 0.17 |
> | SNGP [4] | 72.77 $\pm$ 1.23 | 76.63 $\pm$ 1.51 | 71.91 $\pm$ 6.19 | 73.54 $\pm$ 5.36 | 73.91 $\pm$ 1.85 | 74.53 $\pm$ 1.93 | 76.19 $\pm$ 1.19 |
> | **Ours** | **81.86 $\pm$ 0.19** | **83.40 $\pm$ 0.09** | **86.17 $\pm$ 1.83** | **85.07 $\pm$ 1.94** | 78.58 $\pm$ 0.33 | **79.97 $\pm$ 0.39** | **78.29 $\pm$ 0.11** |
>
> We found that DUQ and DUE fail to converge on CIFAR-100 when using the same settings that work on CIFAR-10. As their original papers do not report CIFAR-100 results, there is little guidance for adapting these methods to more complex datasets.
>
>
> ## Response to Comment 2
> Thank you for your constructive comment. We have added misclassification detection experiments using Excess Area Under the Risk-Coverage curve (E-AURC *1000) as the evaluation metric, which is widely adopted for misclassification detection task. We use conditional entropy as the aleatoric uncertainty measure.
>
> Our method consistently achieves the lowest E-AURC scores across different corruption levels on CIFAR10-C, indicating its strong ability to provide well-calibrated uncertainty that correlates with misclassification likelihood. This further validates the practical utility of our approach beyond OOD detection.
>
> | Method       | s=1 | s=2 | s=3  | s=4  | s=5   | mean |
> | ------------ | --- | --- | --- | --- | --- |  --- |
> | EDL          |  18.12 $\pm$ 0.31 | 30.16 $\pm$ 0.96  |   44.54 $\pm$ 1.95 |  63.61 $\pm$  1.90  |  101.61 $\pm$ 4.46 |      51.60 $\pm$ 1.91
> | RED          |   16.74 $\pm$ 0.40  |   30.15 $\pm$ 1.40   |  44.80 $\pm$ 1.66   |    65.26 $\pm$ 2.61  | 103.87 $\pm$ 5.73    |    52.16 $\pm$ 2.36 |
> | I-EDL        |   14.62 $\pm$ 0.52 |   27.84 $\pm$ 0.65   |  41.70 $\pm$ 1.57   |   59.53 $\pm$ 2.92  |  95.93 $\pm$ 4.43   |      47.92 $\pm$ 2.01 |
> | R-EDL        |  17.13 $\pm$ 0.47  |  29.85 $\pm$ 1.05  |  45.06 $\pm$ 0.74   |   63.75 $\pm$ 0.51  | 101.74 $\pm$ 1.21	    |      51.50 $\pm$ 0.79 |
> | DA-EDL       |   20.61 $\pm$ 2.75  |  35.56 $\pm$ 5.31   |  51.09 $\pm$ 8.38  |   72.29 $\pm$ 11.23  |  112.83 $\pm$ 14.53   |      58.47 $\pm$ 8.44 |
> | **Ours**     | **8.70 $\pm$ 0.35** | **14.80 $\pm$ 0.45** | **21.90 $\pm$ 0.81**|**35.06 $\pm$ 1.81** | **66.52 $\pm$ 7.37** | **29.40 $\pm$ 2.16**|
>
> ## Reponse to Comment 3
> We thank the reviewer for this constructive suggestion.
> To address this, we expanded our evaluation to include ImageNet-200 as the ID dataset and conducted OOD detection experiments on five challenging benchmarks: SSB-Hard, NINCO, iNaturalist, Textures, and OpenImage-O.
> These datasets cover diverse natural image variations and are widely adopted for large-scale OOD evaluation in OpenOOD.
>
> Due to space limitation, for detailed experimental setup and hyperparameter configurations, please kindly refer to Response to Comment 1 of Reviewer Vmcj. Here, we reproduce the main results below:
>
>
> | Method     | →SSB-Hard        | →NINCO           | →iNaturalist     | →Textures        | →OpenImage-O      | Cls Acc         |
> |------------|------------------|------------------|------------------|------------------|-------------------|------------------|
> | KL-PN      | 50.00        | 50.00        | 50.00       | 50.00      | 50.00        | 0.50         |
> | RKL-PN     | 50.00      | 50.00         | 50.00      | 50.00       | 50.00   | 0.50        |
> | EDL        |45.55	| 46.34| 40.91| 36.08 | 52.42	 |2.30|
> | RED        |77.81| 82.25 | 85.42| 80.32 | 83.72 |86.19|
> | I-EDL     |48.50| 52.49	| 52.57 | 54.24 | 49.58 |1.09|
> | R-EDL      |50.88| 53.99 | 41.10	| 53.99 | 55.28 |2.67|
> | Our | **79.80**|  **83.58** | **93.93**| **90.71** | **89.84** |**87.33**|
>
> The results show that most standard EDL methods either fail to converge or perform poorly in this large-scale setting, confirming their limitations beyond toy datasets. In contrast, our method demonstrates consistently superior OOD detection performance, highlighting its robustness and scalability.
> # Response to Questions
> ## Response to Benchmark coverage and misclassification detection experiments
> We appreciate the reviewer’s thoughtful comments. We have addressed both points in our Response to Weaknesses above, please kindly refer to that section for further details.
>
> ## Reponse to terrible performances of some methods in benchmark
> First, the terrible performance of KL-PN and RKL-PN on CIFAR-10 has also been reported in prior works, including I-EDL and R-EDL. We suspect this is primarily due to the use of a regularization strategy that imposes a constraint: Dirichlet strength corresponding to incorrect classes are explicitly required to approach zero (e.g., matching a prior distribution). Such a constraint may result in a collapse of feature representations with exponential activation, as the model over-optimizes toward suppressing incorrect class activations rather than learning expressive features when the number incorrect classes dominate the loss computation (i.e. K-1 $\gg$ 1).
>
> Second, we observe that EDL variants based on the Brier score loss—such as EDL, R-EDL, and DA-EDL perform particularly poorly on CIFAR-100 and ImageNet-200 in terms of classification accuracy. This limitation of the Brier score in large-class settings has also been highlighted in the RED method [6], which pointed out its inefficacy when applied to tasks with a large number of categories.
>
>
> ## Reponse to the question why model would also learn to assign high uncertainty to inputs far from the training samples
>
> We hypothesize that this behavior arises from the distance-preserving property of level-1 vicinal labels, which better reflect the similarity structure of the input space. Unlike level-0 hard labels that enforce sharp one-hot decisions, vicinal level-1 labels encode continuous relationships between nearby samples. Consequently, EDL models trained on such labels tend to learn feature representations that are more sensitive to distances in the input space, naturally leading to higher uncertainty in regions far from the training data manifold.
>
> In contrast, when EDL is trained with level-0 hard labels, we observe that the classifier vectors tend to converge to solutions similar to those of a $K$-class SVM. The regions of high uncertainty are enclosed by negative margins of decision boundaries between different classes of a $K$-clss SVM. This phenomenon, known as the implicit bias of neural networks [5], has attracted increasing attention in recent research and is considered particularly intriguing.
>
>
> ## Response to uncertainty-scoring function in Figure 1
> As clarified in lines 267–270, we define uncertainty as the vacuity of evidence, computed as K ⁄ S for consistency, and this is the measure used in Figure 1 as well.
> We will make this explicit in the caption of Figure 1 for clarity and completeness.
>
>
>
> ### Reference
>
> [1] Uncertainty Estimation Using a Single Deep Deterministic Neural Network. van Amersfoort, Joost and Smith, Lewis and Teh, Yee Whye and Gal, Yarin. ICML 2020.
>
> [2] Deep deterministic uncertainty: A new simple baseline.
> Mukhoti, Jishnu and Kirsch, Andreas and van Amersfoort, Joost and Torr, Philip HS and Gal, Yarin. CVPR 2023.
>
> [3] On Feature Collapse and Deep Kernel Learning for Single Forward Pass Uncertainty. van Amersfoort, Joost and Smith, Lewis and Jesson, Andrew and Key, Oscar and Gal, Yarin.
>
> [4] Simple and principled uncertainty estimation with deterministic deep learning via distance awareness. Liu, Jeremiah and Lin, Zi and Padhy, Shreyas and Tran, Dustin and Bedrax Weiss, Tania and Lakshminarayanan, Balaji. NIPS 2020
>
> [5] Soudry D, Hoffer E, Nacson M S, et al. The implicit bias of gradient descent on separable data[J]. Journal of Machine Learning Research, 2018.
>
> [6] Pandey D S, Yu Q. Learn to accumulate evidence from all training samples: theory and practice[C]//International Conference on Machine Learning. PMLR, 2023.

---

> ### Comment · Area_Chair_iXzU · 2025-08-04
>
> Dear Reviewer,
>
> Please engage in the discussion with the authors. The discussion period will end in a few days.
>
> Thanks,
>
> AC

---

### Official Review · Reviewer_LVLn · 2025-06-30

**Clarity:** 2
**Significance:** 2
**Originality:** 2
**Rating:** 4
**Confidence:** 3

**Summary:**

Evidential Deep Learning (EDL) is a widely used approach for uncertainty quantification (UQ) that models predictive uncertainty via a second-order distribution---specifically, a Dirichlet distribution over the probability simplex in classification tasks. Recent studies have pointed out that most existing EDL methods rely on level-0 supervision (i.e., hard labels), which can lead to unfaithful uncertainty representations. In particular, the learned Dirichlet distribution may become degenerate, collapsing to a Dirac delta function.

This work aims to address this issue by incorporating level-1 supervision, i.e., categorical label distributions. Since such ground-truth level-1 labels are typically expensive to obtain, the authors propose generating them using techniques inspired by MixUp. Two types of level-1 labels are constructed: one to model aleatoric uncertainty and another for epistemic uncertainty. The UQ model is then trained using a vicinal risk minimization (VRM) objective based on the constructed labels. The authors argue that this approach mitigates the degeneracy issue of Dirichlet distributions and enables more faithful uncertainty modeling. Empirical results demonstrate that the proposed method yields improved performance on downstream tasks such as out-of-distribution (OOD) detection.

**Questions:**

1.   Can you elaborate on the sensitivity to hyperparameters?  More generally, it is helpful to understand in what scenarios the method will fail or otherwise not be useful, and how it will fail when it does.

2.  How can the approach of Section 3.2 be justified in light of the comments above?

3.  How would the authors address the comment regarding one of the claims, as mentioned above?

**Ethical Concerns:**

["NO or VERY MINOR ethics concerns only"]

**Final Justification:**

While justifying analysis is limited, empirical benefits have been demonstrated.   Having gone through the responses to my review as well as the other reviews and responses, I am willing to raise my score from a 3 to a 4.

**Limitations:**

Yes.

**Paper Formatting Concerns:**

None.

**Quality:**

2

**Strengths And Weaknesses:**

Strengths:

- This work tackles an important research problem: overcoming the limitations of existing EDL methods that struggle to learn faithful representations of uncertainty.

- The motivation and overall direction are intuitive. Specifically, the paper clearly identifies the key issue, EDL methods learning second-order distributions from level-0 (hard) labels, and proposes constructing level-1 (soft) labels as a principled solution.

- The proposed method demonstrates strong empirical performance on downstream UQ tasks.

Weaknesses:

- The contribution of the paper is somewhat limited, as several of the key arguments have already been well-established in the existing uncertainty quantification (UQ) literature. For example, prior works [2, 3] have pointed out that a core issue with EDL lies in its reliance on level-0 hard labels, which can cause the learned Dirichlet distribution to collapse into a Dirac measure. Additionally, [39] has shown that existing EDL methods fail to faithfully capture both aleatoric and epistemic uncertainty. Furthermore, both [22] and [39] emphasize the importance of incorporating level-1 supervision into the EDL framework, and [39] advocates for distillation-based approaches that construct level-1 labels through ensemble or bootstrap-based techniques (see Section 6 of [39]).

- The proposed method is heuristic and appears to be sensitive to hyperparameters. While level-0 hard labels are sampled from an underlying true distribution, the level-1 labels in this work are constructed heuristically via MixUp. As a result, the distribution of these crafted labels, and by extension, the behavior of the learned UQ model, depends on the specific MixUp parameters. The ablation studies in Section 5 further confirm that the proposed method performs well only under well-tuned hyperparameter settings.

- Some of the paper’s claims are not entirely accurate. Specifically, the concern regarding the Dirac delta collapse applies only to EDL objectives without regularization. In practice, many EDL methods employ regularizers, which prevent the learned Dirichlet from collapsing into a delta function. As shown in [39], with regularization, the learned second-order distribution converges to a fixed-form Dirichlet (see Example 5.2), determined by the regularization hyperparameter. Thus, while the distribution may not collapse, it does not necessarily mean the model has learned faithful or meaningful uncertainty.

- The justification for the proposed approach in Section 3.2 is unconvincing. The authors suggest that mixing in-distribution samples with random noise helps epistemic uncertainty estimation and argue that perturbed samples should exhibit greater epistemic uncertainty than the clean samples. However, this assumption is questionable. Epistemic uncertainty reflects the model's lack of knowledge and should not depend on the input distribution. A simple counterexample illustrates this: consider a human expert acting as a highly confident classifier with negligible epistemic uncertainty. If a clean image is perturbed with sufficient random noise, even a human may become uncertain, not due to lack of knowledge of a human expert (epistemic uncertainty), but because the image has become ambiguous or unrecognizable (aleatoric uncertainty). This suggests that adding noise primarily increases aleatoric uncertainty, not epistemic uncertainty, as the model would struggle to make reliable predictions regardless of additional training data.

---

> ### Author Rebuttal · Authors · 2025-07-30
>
> # Response to Weakness
>
> ## Response to Comment 1
> We sincerely appreciate the reviewer’s careful reading and valuable comments. We fully acknowledge that the core issue of Dirichlet degeneration under level-0 supervision has been identified in prior works such as [2, 3, 22, 39]. However, we would like to clarify several key differences and contributions of our method compared to these existing works:
> 1. Consideration of aleatoric uncertainty:
> While [2, 3, 22] primarily focus on epistemic uncertainty, their methods largely overlook the modeling and improvement of aleatoric uncertainty. Our work explicitly addresses this gap by designing strong MixUp-based vicinal level-1 supervision to provide richer and more effective signals for aleatoric uncertainty estimation, which is often under-explored in existing EDL literature.
>
> 2. Limitations of distillation labels for aleatoric uncertainty estimation:
> While [39] proposes a distillation-based approach to construct level-1 soft labels without requiring ground-truth distributions, we argue that such soft labels are inherently limited in representing aleatoric uncertainty. Specifically, distilled labels are typically:
>
>     - derived from ensemble predictions that tend to average out input ambiguity, and
>     - often remain close to hard labels, failing to reflect the true uncertainty present in ambiguous or borderline samples.
>
>     As a result, we believe distillation is not well-suited for modeling aleatoric uncertainty.
>     In contrast, our method generates level-1 supervision using strong MixUp between semantically distinct examples, explicitly injecting ambiguity and producing soft labels that more faithfully capture aleatoric uncertainty.
>     For instance, although [39] lacks public code for direct comparison, Figure 4 in [39] shows their method underperforms I-EDL on misclassification detection. In our experiments, we surpass I-EDL on this task (see Response to Reviewer XHmk, Comment 2), highlighting the strength of our approach for aleatoric uncertainty estimation.
>
>
> 3. Efficiency and scalability:
> Ensemble-based distillation approaches like [39] require training multiple models, which can be computationally expensive and less practical in large-scale or resource-limited settings. Our method requires training only a single model, making it significantly more efficient and scalable.
>
>
>
> ## Response to Comment 2
> We appreciate the reviewer’s concern regarding the heuristic nature and hyperparameter sensitivity of our proposed method.
> While it is true that our method involves hyperparameters such as $\beta$ and $\beta_{\text{noise}}$, we would like to emphasize two key points:
>
> 1. Theoretical guidance for hyperparameter selection: Unlike existing methods that rely purely on empirical tuning (e.g., KL divergence regularization or entropy constraints often require manually chosen weight coefficients without theoretical justification), our approach is supported by theoretical analysis. Specifically:
>
>
>     - Theorem 2 suggests that using a larger $\beta$ (i.e., $\beta \gg 1/2$) encourages stronger interpolations, which in turn enhance model generalization by mitigating input-dependent label noise introduced by level-0 hard labels. This insight departs from standard MixUp settings (e.g., $\beta = 0.2$) and offers a principled approach to narrowing the hyperparameter search range.
>
>     - Theorem 3 further links the choice of $\beta_{\text{noise}}$ to the statistical behavior of sampled $\lambda$ values in the noise-based mixup. It highlights that overly large $\beta_{\text{noise}}^+$ leads to $\lambda \approx 1$, which may lead to an excessively rapid increase in Dirichlet strength.
>     Thus, our theory also guides the selection of $\beta_{\text{noise}}^+$ to avoid such degenerate cases.Together, these theoretical results significantly reduce the burden of hyperparameter tuning by informing practitioners what regimes to avoid and where optimal values are likely to lie.
>
>
> 2. We also point out that existing EDL methods are not free from hyperparameter sensitivity. For example, methods that incorporate KL divergence or entropy constraints often require selecting an appropriate regularization weight (e.g., $\lambda_{\text{KL}}$), yet such choices are entirely empirical and lack theoretical grounding. In contrast, our method provides both practical effectiveness and theoretical explainability.
>
>
> ## Response to Comment 3
> We thank the reviewer for pointing out that Dirac delta collapse primarily arises in EDL without regularization.
> We completely agree with this point, and indeed, our Theorem 1 is based on the same observation.
> At the same time, we would like to clarify that our paper does not claim that non-degenerate Dirichlet distributions (e.g., those obtained via regularization) necessarily lead to reliable uncertainty estimation. In this regard, we fully share the reviewer’s perspective.
> However, our work focuses on a related but distinct issue: models trained solely with level-0 labels often fail to learn faithful or meaningful uncertainty estimates.
> In other words, avoiding collapse is necessary but not sufficient for achieving well-calibrated uncertainty. This insight underpins our motivation to incorporate level-1 supervision.
> While our work does not directly investigate regularization-based EDL objectives, we believe we share the reviewer’s perspective on their limitations. In particular, many commonly used regularizers (e.g., entropy-based penalties) primarily control the overall spread or shape of the Dirichlet distribution, without necessarily aligning the uncertainty estimates with the underlying data ambiguity or model ignorance.
> As a result, such regularization may help prevent degenerate solutions but still fail to produce reliable or calibrated uncertainty estimates, especially in regions of input space that are ambiguous or underrepresented. This further motivates our use of level-1 supervision, which provides more targeted guidance for modeling both aleatoric and epistemic uncertainty.
> We appreciate the reviewer’s clarification and will revise the manuscript to better position our work relative to regularization-based methods and to make it explicit that we do not claim regularization alone ensures reliable uncertainty quantification.
>
>
> ## Response to Comment 4
> We thank the reviewer for the insightful comment on Section 3.2. We would like to clarify that aleatoric uncertainty arises from inherent  semantic ambiguity—such as when multiple annotators assign different labels to a perceptually ambiguous image. In contrast, our method does not introduce label randomness. Even when the input is perturbed with noise, we assign a consistent ground-truth label to each generated sample, and thus do not increase label uncertainty.
> Moreover, the Gaussian noise introduced in Section 3.2 is milder compared to the strong MixUp strategy discussed in Section 3.1, and therefore does not lead to a higher level of aleatoric uncertainty.
> The objective of Section 3.2 is to simulate scenarios where the input becomes ambiguous or unfamiliar due to perturbations. By applying smoothed level-0 labels, we avoid excessive Dirichlet concentration, following the principle that epistemic uncertainty is closely tied to the dispersion of the Dirichlet distribution.
> In summary, our method does not conflate aleatoric and epistemic uncertainty. Instead, it leverages fixed supervision to guide the model toward expressing lower confidence (i.e., higher epistemic uncertainty) when encountering inputs that lie off the data manifold. We will revise Section 3.2 to make this distinction clearer in the final version.
>
>
>
> # Response to Questions
>
> ## Response to Question 1
> Thank you for raising this important point regarding the sensitivity to hyperparameters and potential failure cases of our method.
> To provide a more thorough understanding of the robustness of our approach, we kindly invite the reviewer to refer to our response to Comment 2 from Reviewer 9dhq, where we independently activated each loss component to assess its individual contribution. This analysis indicates that, although the method does benefit from appropriate tuning, the effective range of hyperparameters aligns well with the theoretical guidance provided by Theorems 2 and 3.
>
> ## Response to Question 2
>
> We thank the reviewer for raising this important question regarding the justification of our approach in Section 3.2. The key intuition behind our method is to approximate epistemic uncertainty via a smooth supervision signal, inspired by label smoothing and vicinal risk minimization principles.
> Specifically, we perturb in-distribution samples with increasing levels of noise, and assign increasingly smooth target distributions (i.e., closer to uniform) as supervision. This construction is based on the observation that such perturbed inputs are more likely to lie away from the training data manifold, and hence the model should express lower confidence (higher epistemic uncertainty) in its predictions.
> Importantly, we emphasize that our method does not inject label noise—the supervision remains controlled and structured. Rather than introducing randomness, we deliberately reduce the confidence on the original label as the input becomes less familiar to the model. This aligns with the core idea of epistemic uncertainty: uncertainty due to limited knowledge or lack of training coverage.
> While our approach is heuristic in nature, we find it empirically effective for improving model behavior on OOD and ambiguous inputs. We will revise Section 3.2 to clarify this motivation and better distinguish between label smoothing for epistemic modeling versus aleatoric uncertainty due to intrinsic ambiguity.
>
>
> ## Response to Question 3
>  Thank you for your question. Please kindly refer to our previous comments.

---

> > ### Comment · Reviewer_LVLn · 2025-08-04
> > **comments on response**
> >
> > Most of my concerns have been addressed, and I encourage the authors to explicitly acknowledge the limitations of their approach in the final version. However, I remain unconvinced by the core assumption in Section 3.2 that perturbing in-distribution samples with noise can reliably simulate epistemic uncertainty. Epistemic uncertainty arises from a model’s lack of knowledge, typically due to insufficient training data. Simply injecting noise into in-distribution samples does not necessarily create this kind of knowledge gap. First, consider the "human expert" example mentioned in my review comments: if different experts are asked to label a heavily noised image, their disagreement likely arises because the image is ambiguous, not because human lack knowledge about the classifying an image. Second, for neural networks, adding noise does not guarantee that the resulting sample lies outside the training distribution. The perturbed image may still fall within a high-density, aleatoric region of the underlying data manifold. In such cases, the observed uncertainty does not reflect epistemic ignorance but rather inherent input ambiguity.    A further response from the authors would be helpful.

---

> > > ### Author Response · Authors · 2025-08-05
> > >
> > > We sincerely thank the reviewer for the continued discussion and insightful comments. Epistemic uncertainty—stemming from the model’s lack of knowledge—is inherently challenging to model in practice. Previous approaches such as KL-PriorNet and Reverse-KL PriorNet address epistemic uncertainty from both the input and output perspectives. Specifically, they introduce inputs from explicitly out-of-distribution (OOD) regions to represent the model’s knowledge gap on the input side, while assigning a uniform Dirichlet distribution as the supervision target on the output side to encourage maximal uncertainty. However, in real-world settings, it is often impractical to obtain well-defined OOD inputs that reliably reflect such knowledge gaps.
> > >
> > > As the reviewer rightly noted, our method does not attempt to rigorously construct knowledge-gap inputs. We believe that explicitly generating inputs that exhibit a clear knowledge gap relative to the in-distribution data is an ill-posed problem, as it is fundamentally difficult to formalize what constitutes an unknown region of the input space.
> > > Instead, we focus on the output side. By smoothing the supervision signal, we effectively slow the growth of Dirichlet strength, preventing the model from collapsing its predictive distribution into overly confident (Dirac-like) forms in ambiguous  regions. While this strategy does not fully address the knowledge gap in the input space, it aims to enhance epistemic uncertainty estimation by preserving a meaningful level-2 predictive distribution in uncertain cases.
> > >
> > > Importantly, the main contribution of our work lies in improving the modeling of aleatoric uncertainty, which prior methods have largely overlooked. This is validated by performance gains in misclassification detection tasks. In addition, our approach contributes to the broader goal of modeling epistemic uncertainty through output-space belief calibration, even though the input-space perspective remains only indirectly addressed.
> > > We will revise Section 3.2 to more clearly articulate these motivations and limitations, and to better position our method as a step toward more comprehensive epistemic modeling.
> > >
> > > Once again, we sincerely thank the reviewer for their thoughtful feedback, and we would greatly appreciate the opportunity to continue engaging with you and receiving your valuable guidance.

---

> > > > ### Comment · Reviewer_LVLn · 2025-08-06
> > > > **concluding comment**
> > > >
> > > > Thank you for the response.   The analysis is limited, and there doesn't appear to be any sharp guarantee that proposed method disentangles epistemic and aleatoric uncertainty simply by adding noise. Nevertheless, such heuristic approaches do appear to show empirical benefits on downstream tasks.
> > > >
> > > > Based on this, and having gone through the other reviews and responses as well, I am willing to raise my score from a 3 to a 4.

---

> > > > > ### Author Response · Authors · 2025-08-07
> > > > >
> > > > > Thank you very much for your careful reconsideration and for raising your score to a 4. We greatly appreciate your insightful observation. The  feedback will guide us to further explore both the theoretical foundations and practical implications of our method in future work. We are grateful for your time and constructive input.

---

### Official Review · Reviewer_9dhq · 2025-07-02

**Clarity:** 4
**Significance:** 3
**Originality:** 3
**Rating:** 5
**Confidence:** 5

**Summary:**

Evidential Deep Learning aims to estimate both aleatoric and epistemic uncertainty using Dirichlet distributions, but existing approaches relying on hard labels (level-0 supervision) often produce degenerate outputs with unreliable uncertainty estimates, including both aleatoric and epistemic uncertainty. This paper proposes a vicinal risk minimization framework that constructs level-1 supervisory signals through local label interpolation, without requiring additional annotations. The authors provide a theoretical analysis showing improved generalization by reducing the lower bound of generalization error, and empirical results demonstrate consistent gains over baselines in out-of-distribution detection and generalization tasks.

**Questions:**

1. The proposed method builds on mixup and label smoothing strategies, and Table 3 shows it improves ID accuracy even for MSP. How can we tell whether the performance gains in Tables 1 and 2 come from improved representation learning versus improved uncertainty modeling? For example, can we analyze latent representations (e.g., in PostNet) before and after applying the proposed method?

2. The paper introduces two new uncertainty quantification measures—Dirichlet differential entropy and conditional entropy. What is the motivation for using these specific measures, and is there any experimental evidence showing their advantages over existing metrics?

3. How does Theorem 2 theoretically support improved generalization and robustness? Does it also suggest an improvement in the quality of the uncertainty estimates?

**Ethical Concerns:**

["NO or VERY MINOR ethics concerns only"]

**Final Justification:**

EDL methods offer valuable explainability through multi-dimensional uncertainty estimates, but are known to produce unfaithful uncertainty estimation. Learning a faithful second-order distribution remains a challenging and active research topic. While this paper can not fully resolve the collapsing problem in EDL, it may provide useful ideas for further exploration of EDL methods. Therefore, I maintain my score.

**Limitations:**

yes

**Quality:**

3

**Strengths And Weaknesses:**

**Strengths**
1. The paper is well-written and easy to follow; the motivation is clearly articulated, the methodology is well structured, and the appendix provides detailed derivations and theoretical proofs.
2. The experimental evaluation includes a broad set of EDL-related baselines across both out-of-distribution (OOD) detection and OOD generalization tasks, showing consistent improvements of the proposed method.
3. The proposed approach is simple to implement and can be readily integrated into other uncertainty-aware models and tasks without requiring additional annotation or architectural changes.

**Weaknesses**
1. The paper claims that standard EDL methods fail to capture reliable aleatoric uncertainty and that the proposed method improves this; however, the evaluation would benefit from more targeted experiments measuring aleatoric uncertainty quality, such as misclassification detection or selective prediction.

2. The total loss consists of two components (Equation 14), i.e., vicinal supervision and noise augmentation. However, the paper lacks an ablation study isolating their individual contributions, which would help clarify the effectiveness of each.

3. Since the proposed method involves mixup and additional data generation, a brief analysis or discussion of computational complexity and training overhead would strengthen the practical understanding of the method's scalability.

---

> ### Author Rebuttal · Authors · 2025-07-30
>
> # Response to Weakness
> ## Response to Comment 1
> Thank you for highlighting the need for more targeted evaluations of aleatoric uncertainty. In response, we conduct selective prediction experiments on CIFAR-10-C using the Excess Area Under the Risk-Coverage Curve (E-AURC × 1000, lower is better), a standard metric for misclassification detection task. We adopt conditional entropy to measure aleatoric uncertainty.
> Our method consistently achieves the lowest E-AURC across all severities, demonstrating more reliable aleatoric uncertainty estimation and improved performance in selective prediction scenarios.
> We will include these results in the revised manuscript.
>
> | Method       | s=1   | s=2  | s=3   | s=4   | s=5   | mean  |
> | ------------ | --- | --- | --- | --- | --- |  --- |
> | EDL          |  18.12 $\pm$ 0.31 | 30.16 $\pm$ 0.96  |   44.54 $\pm$ 1.95 |  63.61 $\pm$  1.90  |  101.61 $\pm$ 4.46 |      51.60 $\pm$ 1.91
> | RED          |   16.74 $\pm$ 0.40  |   30.15 $\pm$ 1.40   |  44.80 $\pm$ 1.66   |    65.26 $\pm$ 2.61  | 103.87 $\pm$ 5.73    |    52.16 $\pm$ 2.36 |
> | I-EDL        |   14.62 $\pm$ 0.52 |   27.84 $\pm$ 0.65   |  41.70 $\pm$ 1.57   |   59.53 $\pm$ 2.92  |  95.93 $\pm$ 4.43   |      47.92 $\pm$ 2.01 |
> | R-EDL        |  17.13 $\pm$ 0.47  |  29.85 $\pm$ 1.05  |  45.06 $\pm$ 0.74   |   63.75 $\pm$ 0.51  | 101.74 $\pm$ 1.21	    |      51.50 $\pm$ 0.79 |
> | DA-EDL       |   20.61 $\pm$ 2.75  |  35.56 $\pm$ 5.31   |  51.09 $\pm$ 8.38  |   72.29 $\pm$ 11.23  |  112.83 $\pm$ 14.53   |      58.47 $\pm$ 8.44 |
> | **Ours**     | **8.70 $\pm$ 0.35** | **14.80 $\pm$ 0.45** | **21.90 $\pm$ 0.81**|**35.06 $\pm$ 1.81** | **66.52 $\pm$ 7.37** | **29.40 $\pm$ 2.16**|
>
> ## Response to Comment 2
> Thank you for the valuable suggestion. We agree that isolating the contributions of the vicinal supervision and noise augmentation losses would provide a clearer understanding of their individual effects.
> To address this, we have conducted an ablation study to individually analyze the contributions of the two components in the total loss.
> First, to isolate the effect of $ \beta $, we first remove the loss term  $ L_{noise} $  and focus solely on its influence.
>
> | $\beta$ | $\beta_{\text{noise}}^+$| MisD E-AURC ↓ | OOD AUROC ↑ | OOD Acc ↑ | ID Acc |
> |--------:|--------:|-----------------------------:|------------:|----------:|---------:|
> |-| -            |        56.54   $\pm$ 3.98   |           90.67 $\pm$ 0.35    |      74.51   $\pm$ 0.49       | 95.17  $\pm$ 0.18 |
> | 0.2     |   -  |          58.10   $\pm$ 6.04             |     89.77 $\pm$ 0.63        |     76.39 $\pm$ 0.55      | 95.31 $\pm$ 0.08|
> | 0.4     |  -    |        49.44 $\pm$ 4.40                |    90.28 $\pm$ 0.05          |   77.82 $\pm$ 0.61        | 95.61 $\pm$ 0.16|
> | 1.0     |   -     |         45.18 $\pm$ 2.19          |   91.11 $\pm$ 0.40              |     79.13 $\pm$ 0.32      | 96.01 $\pm$ 0.02|
> | 5.0     |   -      |      42.94 $\pm$ 4.37               |       91.68 $\pm$ 0.48      |    79.75 $\pm$ 1.02       | 95.89 $\pm$ 0.21 |
> | 10.0    |   -       |    39.15 $\pm$ 4.33                |   91.73 $\pm$ 0.14          |   80.41 $\pm$ 1.23        | 95.86 $\pm$ 0.16 |
>
> We highlight several key observations from our ablation study:
> 1. Improved Aleatoric Uncertainty Estimation for Misclassification Detection:
> Increasing $\beta$ leads to a monotonic decrease in E-AURC, indicating better calibration of aleatoric uncertainty for misclassification detection. E-AURC drops significantly from 56.54 (baseline) to 39.15 at $\beta = 10.0$.
>
> 2. Enhanced OOD Detection Performance:
> Although our primary goal is to improve aleatoric uncertainty estimation with $ L_{vicinal} $ , we observe consistent gains in both OOD AUROC as $\beta$ increases. This suggests that vicinal supervision not only calibrates aleatoric uncertainty but also enhances the model’s discriminative ability between ID and OOD samples. We attribute this to the regularizing effect of vicinal labels, which expose the model to local variations around the data manifold and thus improve generalization beyond the training distribution—even in the absence of $\mathcal{L}_{\text{noise}}$.
>
> 3. Enhanced Generalization Performance:
> We also observe simultaneous improvements in both OOD classification accuracy and ID accuracy as $\beta$ increases. Specifically, OOD accuracy rises from 74.51% to 80.41%, while ID accuracy improves from 95.17% to 95.86%. These gains follows Theorem 2 that strong mixup strengthens the model’s overall generalization across both known and unseen data distributions.
>
> We further analyze the impact of $\beta_{noise}^+$ by isolating the loss to $L = L_{noise}$. Following the setup in our main paper, we fix $\beta_{noise}^- = 1.0$ and vary $\beta_{noise}^+$.
>
> |$\beta$| $\beta_{\text{noise}}^+$ | OOD AUROC (↑) | MisD E-AURC   (↓) | OOD Acc  (↑) | ID Acc  (↑)|
> |------|----------------------------|-----------|-------------|--------------|--------------|
> |-| -            |     90.67  $\pm$ 0.35      |     56.54   $\pm$ 3.98       |      74.51   $\pm$ 0.49     | 95.17 $\pm$ 0.18|
> |-| 0.2                        |   90.95 $\pm$ 0.32       |   56.11 $\pm$ 4.92        |  74.32 $\pm$  0.43            | 95.00 $\pm$ 0.02  |
> |-| 0.4                        |   91.18 $\pm$ 0.42      |      62.81 $\pm$ 10.15       |       73.76 $\pm$ 0.67     | 95.03 $\pm$ 0.21 |
> |-| 1.0                        |   91.85 $\pm$ 0.24        |     57.22 $\pm$ 6.14        |     73.96 $\pm$ 0.43         | 95.17 $\pm$ 0.09  |
> |-| 5.0                        |    90.74 $\pm$ 0.20       |     55.95 $\pm$ 6.40        |     74.67 $\pm$ 0.75         | 95.02 $\pm$ 0.20 |
> |-| 10.0                       |        90.37 $\pm$ 0.55    |    60.17 $\pm$ 2.85       |      73.42 $\pm$ 0.53        | 94.86 $\pm$ 0.41 |
> | 10.0  |          1.0         |    **93.08 $\pm$ 0.33**       |   **29.40 $\pm$ 2.16**             |      **88.73 $\pm$ 0.13**        | **96.18 $\pm$ 0.13** |
>
> We highlight several key observations:
> 1. Isolated Variation of $\beta_{\text{noise}}^+$ Alone Has Limited Impact on MisD and Cls Acc:
> When $L_{vicinal}$ is removed and only $\beta_{\text{noise}}^+$ varies (rows with “-” under $\beta$), E-AURC and ID Acc fluctuate slightly but do not show consistent improvement,  indicating that tuning $\beta_{\text{noise}}^+$ alone without adjusting $\beta$ with vicinal information may be insufficient for substantial gains.
>
> 2. Balancing Noise Intensity Is Crucial:
> Setting $\beta_{\text{noise}}^+$ too small can impair classification accuracy, as overly aggressive noise perturbations dominate the input and hinder effective learning. Conversely, a very large $\beta_{\text{noise}}^+$ causes the sampled mixing coefficient $\lambda$ to concentrate near 1, effectively disabling noise augmentation and reverting the method back to standard EDL behavior. In contrast, $\beta_{\text{noise}}^+ = 1.0$ achieves the best trade-off for epistemic uncertainty estimation and OOD detection.
>
> 3. **Best Performance Achieved with Combined $\beta = 10.0$ and $\beta_{\text{noise}}^+ = 1.0$:**
>  The model attains the highest **OOD AUROC (93.08%)**, lowest **MisD E-AURC (29.40)**, best **OOD accuracy (88.73%)**, and highest **ID accuracy (96.18%)** when both parameters are optimized together. This highlights the **synergistic effect** of these two components.
>
>
> ## Response to Comment 3
> Our method introduces minimal overhead from mixup, as it only involves simple linear interpolation. The main cost comes from extra forward and backward passes, roughly doubling training time compared to standard EDL. Importantly, our method adds no overhead at inference, keeping test-time cost unchanged.
>
> # Response to Questions
> ### Response to Question 1
> To further investigate whether the improvements in classification accuracy and uncertainty estimation stem from enhanced feature representations, we conducted a feature visualization analysis using t-SNE. We exactly observed that our method with vicinal labels learns more distinct boundaries between ID and semantic-shift OOD features, leading to more reliable epistemic uncertainty estimates. Additionally, it yields more robust feature representations for both covariate-shift OOD samples and ID samples, contributing to improved classification performance on both ID and OOD data. We will include these feature visualizations in a future revision of the paper.
> ### Response to Question 2
> Dirichlet differential entropy reflects both the concentration and dispersion of the distribution, while conditional entropy more directly corresponds to aleatoric uncertainty, as it measures the expected entropy of the predictive categorical distribution [1].
> Empirically, we find that entropy-based metrics are particularly effective for models trained with level-1 supervision. We will include a comparison against other uncertainty measures such as vacuity of evidence and mutual information in the updated appendix.
>
> ### Response to Question 3
>  Theorem 2 rigorously proves that strong MixUp effectively reduces sample-dependent label noise introduced during training. Specifically, the conventional level-0 hard labels act as supervision signals that inject overconfident noise, which lowers the signal-to-noise ratio between the true label information and noise inherent in each sample. This often causes the model to overfit near class boundaries, negatively impacting generalization and robustness.
> By constructing vicinal samples through strong MixUp, the strength of this overconfident noise is effectively diluted, enhancing the purity and consistency of label signals. As a result, the model achieves improved uncertainty estimation capability. Therefore, Theorem 2 not only theoretically supports improved generalization and robustness, but also suggests an enhancement in the quality of uncertainty quantification.
>
> [1] Malinin A, Gales M. Predictive uncertainty estimation via prior networks[J]. Advances in neural information processing systems, 2018, 31.

---

> > ### Comment · Reviewer_9dhq · 2025-08-04
> >
> > Thank the authors for the detailed response and the additional experiments. I do not have further comments.

---

> > > ### Author Response · Authors · 2025-08-05
> > >
> > > We sincerely thank the reviewer for their time, constructive feedback, and kind conclusion. We greatly appreciate your efforts in reviewing our work.

---

### Official Review · Reviewer_Vmcj · 2025-07-03

**Clarity:** 3
**Significance:** 3
**Originality:** 3
**Rating:** 5
**Confidence:** 4

**Summary:**

The paper proposes to further improve the epistemic and aleatoric uncertainty estimation using vicinal risk minimization for evidential deep learning. The paper provides a three-level view: level-0 being hard labels, level-1 being categorical distributions, level-2 being Dirichlet distributions. The paper analyzed the limitations of EDL (level-2 learning against level-0 labels) and proposes to use vicinal supervision to enhance the aleatoric uncertainty estimation and noise-augmented vicinal risk minimization for epistemic uncertainty estimation. The proposed method is used for OOD detection and generalization on several common image datasets.

**Questions:**

1. How realistic are the assumptions in Theorem 1?

2. Can the analysis in Theorem 3 be generalized to other types of evidential losses?

3. Is there more analysis on the ablation study in Figure 3?

**Ethical Concerns:**

["NO or VERY MINOR ethics concerns only"]

**Final Justification:**

I think the paper provides a novel improvement to EDL with some theoretical justification, which should be considered as a useful contribution to the uncertainty quantification and OOD detection/generalization tasks in ML. I agree with Reviewer 9dhq that the proposed method somewhat depends on the hyperparameter and hope that the revised version can better analyze the ablation study part and provide more comprehensive results (specific datasets, detailed guidelines from rebuttal). Overall, I maintain a positive opinion about the paper.

**Limitations:**

The limitation of datasets and tasks is briefly mentioned, but could be better discussed.

**Paper Formatting Concerns:**

The color map and tick font size in Figure 1 can be improved.

**Quality:**

3

**Strengths And Weaknesses:**

Strengths:

1. The paper is well written and most information is clearly presented.

2. Without checking all mathematical details, judging from the high level propositions and conclusions of the theoretical analysis, the theorems are of significance for the uncertainty estimation problem. The theoretical analysis is also essential to the proposed method, showing good consistency.

The main novelty with the theoretical framework is the detailed analysis on learning from level-0/level-1 supervision with level-2 models.  Although level-2 models have already been studied by many (e.g. [3],[2] from the references), Theorem 1 and Proposition 1 formally identifies the insufficiency of traditional risk minimization of level-2 models that also includes the aleatoric uncertainty perspective (rather than epstemic uncertainty alone). To my knowledge, this is a novel contribution and reveals a meaningful characteristic of evidential learning.

The proposed method is not limited to a specific learning task, and the difference comes from the supervision side of things (X and Y). This makes the solution more applicable to various problems, and aligns with the trend of introducing noise into machine learning. This also shows the advantage of VRM in the uncertainty quantification aspect, in addition to generalization.

3. The experimental results show good OOD detection and generalization performances. A sufficient number of recent EDL baselines are included in the comparison.

Weaknesses:

1. As mentioned in the paper, the evaluation is limited to CIFAR scale datasets and the OOD behavior on more realistic tasks is not clear.

2. In the hyperparameter ablation study, the accuracy trend is not clear and the analysis is limited.

3. The notation of $\mathbb{P}_1$ and $\mathbb{P}$ from lines 75-77 can be confusing.

---

> ### Author Rebuttal · Authors · 2025-07-30
>
> # Response to Weakness
>
> ##  Response to Comment 1
> We sincerely appreciate the reviewer’s suggestion regarding the need for evaluation on more realistic large-scale datasets. In response, we conducted additional experiments on the ImageNet-200 dataset, which consists of 200 categories sampled from the full ImageNet-1K. The images are of size 224×224, consistent with the standard ImageNet-1K setting.
> To evaluate the reliability of epistemic estimation in large-scale settings, we first focused on OOD detection tasks where ImageNet-200 serves as the in-distribution (ID) dataset, and SSB-Hard, NINCO, iNaturalist, Textures, and OpenImage-O are used as OOD datasets. We report both in-distribution classification accuracy and OOD AUROC as evaluation metrics.
> The results show that our approach significantly outperforms prior EDL-based methods across all metrics. In particular, models employing KL-PN, RKL-PN, or Brier score-based losses—such as EDL, R-EDL, DA-EDL, and I-EDL—suffered from poor convergence and suboptimal performance. This phenomenon, previously observed on CIFAR-100, becomes even more pronounced on ImageNet-200.
> As noted in the previous work RED, Brier score-based losses tend to struggle with maintaining high classification accuracy when the number of categories is large. RED addresses this issue by introducing evidence regularization to improve class discrimination.
> Our method, by contrast, adopts the expected cross-entropy loss, which avoids the accuracy degradation seen in Brier-based methods. Furthermore, by leveraging vicinal label supervision, our model learns more reliable uncertainty estimates while preserving high classification performance.
>
> In addition, we also include a misclassification detection experiment to evaluate the reliability of aleatoric uncertainty estimation. Specifically, samples that are misclassified should exhibit higher aleatoric uncertainty, while correctly classified samples should have lower uncertainty.
> We use the Excess Area Under the Risk-Coverage Curve (E-AURC × 1000, lower is better), a standard metric for misclassification detection task.
> We find that our method achieves not only higher classification accuracy but also a lower E-AURC, indicating better aleatoric uncertainty calibration.
>
>     Experimental settings:
>     Network: ResNet-18
>     Optimizer: SGD
>     learning rate:0.1
>     lr_scheduler: cosine annealing schedule
>     epochs: 100
>     KL regularization weight for EDL RED I-EDL R-EDL: 0.001 (as suggested in RED for large-scale dataset)
>     Fisher information regularization weight for I-EDL: 0.05
>     Hyperparamter for our method:  $ \beta=10 $, $\beta_noise^+=2$
>
>
>
> | Method     | →SSB-Hard   (OOD AUROC)     | →NINCO   (OOD AUROC)           | →iNaturalist   (OOD AUROC)     | →Textures    (OOD AUROC)      | →OpenImage-O    (OOD AUROC)    | Cls Acc         | MisD E-AURC|
> |------------|------------------|------------------|------------------|------------------|-------------------|------------------| ------------------|
> | KL-PN      | 50.00        | 50.00        | 50.00       | 50.00      | 50.00        | 0.50         | 48.75 |
> | RKL-PN     | 50.00      | 50.00         | 50.00      | 50.00       | 50.00   | 0.50        | 50.25  |
> | EDL        |45.55	| 46.34| 40.91| 36.08 | 52.42	 |2.30|  49.91 |
> | RED        |77.81| 82.25 | 85.42| 80.32 | 83.72 |86.19| 22.62 |
> | I-EDL     |48.50| 52.49	| 52.57 | 54.24 | 49.58 |1.09| 55.98 |
> | R-EDL      |50.88| 53.99 | 41.10	| 53.99 | 55.28 |2.67| 53.60 |
> | Our | **79.80**|  **83.58** | **93.93**| **90.71** | **89.84** |**87.33**| **14.65**  |
>
>
>
>
> ## Response to Comment 2
> Thank you for the insightful comment. We agree that the classification accuracy trend in the original hyperparameter ablation study was not very clear.
> With further ablation study, we found that the classification accuracy is relatively insensitive to variations in these hyperparameters when the other one is held fixed.
> To provide a more thorough analysis, we have added additional experiments where we isolate each loss component and vary the corresponding hyperparameter.
> Please kindly refer to our detailed discussion in the response to Comment 2 from Reviewer 9dhq for further ablation study on hyperparameters.
>
>
>
>
>
> ## Response: Comment 3
> Thank you for the suggestion. We acknowledge that the notation of $\mathbb{P}_1$ and $\mathbb{P}$ could be confusing. We will revise the corresponding expressions in the paper to make the distinction clearer.
>
>
>
>
>
> # Response to Questions
>
> ## Respoonse to Question 1
> The assumptions in Theorem 1 are partially idealized, but aligned with standard practice (level-0 labels + convex level-1 losses). The Dirac assumption enables a clear theoretical analysis of uncertainty collapse. We empirically validate this behavior: the variance of the sampled level-1 distributions $ p \sim \text{Dir}(\alpha) $ is typically less than $10^{-5}$, indicating that level-2 predictions degenerate into Dirac measures $ \delta_p $.
> Furthermore, we observe that the expected predictive distribution over i.i.d. test samples exhibits a sharp peak near 1.0 with a thin tail toward 0, implying that the model assigns nearly all probability mass to a single class. This behavior is consistent with a level-1 Dirac distribution $ \delta_y $, suggesting that the learned uncertainty representation collapses into a single-point prediction, as our theorem characterizes.
>
>
> ## Response to Question 2
> Yes, the analysis in Theorem 3 can be generalized to other types of evidential losses.
> For example, the negative log-likelihood (NLL) loss takes the form
> For log likelihood  takes the form
> $$
> L_{nll}=\sum_{j=1}^K y_j \left( \log (S) - \log (\alpha_j) \right)
> $$
> which can be easily shown to satisfy the conditions in Theorem 3. This is because the derivative of the logarithm function,
> $1/x$, shares similar mathematical properties with the trigamma function $\psi_1(\cdot)$—both are strictly decreasing and strictly convex—making the theoretical analysis straightforward to extend.
> On the other hand, the Brier score loss is given by
>  $$  L_{Brier} = L_{ j}^{ {bias }} + L_{ j}^{ {var }} = \sum_{j=1}^K  \left(y_{ j}-\alpha_{ j} / S\right)^2+ \frac{\alpha_{j}\left(S-\alpha_{j}\right)}{S^2\left(S+1\right)}  $$
> where the variance term
> $$L_{ j}^{ {var }}=\frac{\alpha_{j}\left(S-\alpha_{j}\right)}{S^2\left(S+1\right)} $$
> is independent of the label and reaches its minimum only as
> 𝑆→∞, thereby forcing the Dirichlet distribution toward a degenerate (Dirac) distribution. This behavior is problematic because the minimization pressure is label-agnostic—i.e., even if we use softened labels (e.g., from noise augmentation supervision), this term still encourages overconfident (sharp Dirichlet) predictions regardless of the actual supervision signal.
> Additionally, we include experiments on two types of evidential losses to support our theoretical insights. Specifically, we evaluate our method under both Brier Score and negative log-likelihood losses, comparing models trained with and without vicinal label supervision. To ensure a fair comparison, we exclude the KL regularization term in all settings.
> Experimental results show that, while Brier Score does benefit from vicinal label supervision—particularly in tasks that rely on epistemic uncertainty estimation, such as OOD detection—the improvement in AUROC is noticeably smaller compared to that achieved with log-likelihood loss. This observation aligns well with our earlier analysis: due to the label-agnostic nature of the variance term in Brier Score, its effectiveness in leveraging uncertainty signals is inherently limited.
>
>
> | Loss Type            | Method Used     | ID Acc ↑   | OOD AUROC ↑ |
> |--------------|-----------------|---------|-------------|
> | Brier Score          | w/o Vic   |   94.83 $\pm$ 0.38      |   89.95 $\pm$ 0.29          |
> |  Brier Score                    | w/ Vic   |    95.89 $\pm$ 0.29      |        91.70 $\pm$ 0.26     |
> | Log Likelihood       | w/o Vic   |     94.97 $\pm$ 0.03    |       90.86 $\pm$ 0.18      |
> |     Log Likelihood                 | w/ Vic  |  96.05 $\pm$ 0.13       |    92.8 $\pm$ 0.28         |
>
> ## Response to Question 3
> Thank you very much for your question. Due to space limitations, we were unable to include a more comprehensive analysis in the main paper. Please kindly refer to our detailed discussion in the response to Comment 2 from Reviewer 9dhq for further ablation study on hyperparameters.

---

> > ### Comment · Reviewer_Vmcj · 2025-08-05
> >
> > Thank you for the response. Most of my questions are addressed. I do agree with other Reviewers on some experiment related questions, such as the baseline performances and ablations. I also think that the authors provided adequate responses. I think the main theoretical contribution is novel but I will further review the discussion on certain claims.

---

> > > ### Author Response · Authors · 2025-08-06
> > >
> > > Thank you for your feedback and valuable comments. We are gratified to know that our responses have addressed most of your concerns and we appreciate your recognition of the novelty of our theoretical contribution. With respect to the baseline performance and ablations, we will ensure these are included thoroughly in the revised manuscript. Thank you again for your valuable suggestions and we look forward to receiving further reviews and feedback.

---

> ### Comment · Area_Chair_iXzU · 2025-08-04
>
> Dear Reviewer,
>
> Please engage in the discussion with the authors. The discussion period will end in a few days.
>
> Thanks,
>
> AC

---

### Note · Authors · 2025-08-12

We sincerely thank all reviewers for their thoughtful and constructive feedback, which has been invaluable in further strengthening our work.

**Strengths highlighted by reviewers:**
- Clear and well-motivated problem formulation, with an intuitive overall direction (LVLn, 9dhq).
- Simple to implement, adding minimal complexity beyond standard EDL algorithms while remaining broadly applicable (9dhq, XHmk).
- Well written and clearly presented, making the ideas easy to follow (Vmcj, 9dhq).
- Theoretical significance in extending the scope of uncertainty estimation by explicitly incorporating the aleatoric perspective alongside epistemic uncertainty (Vmcj, XHmk, 9dhq).
- Strong empirical performance across multiple downstream UQ tasks, including better OOD detection and improved generalization (Vmcj, 9dhq, LVLn).

**Key improvements during the rebuttal:**
1. Added *selective classification* experiments, achieving the best results among all compared methods, further validating the effectiveness of our approach for aleatoric uncertainty estimation (9dhq, XHmk).
2. Expanded *outlier detection* benchmarks by incorporating additional baseline methods and scaling evaluations up to ImageNet-level datasets (XHmk, Vmcj).
3. Conducted more extensive ablation studies to isolate and analyze the effects of the two hyperparameters, providing a clearer understanding of their roles (Vmcj, 9dhq, LVLn).
4.  Addressed most of the concerns raised by Vmcj and LVLn, and we sincerely appreciate LVLn for increasing their score from 3 to 4.

**Limitations acknowledged:**
- As LVLn noted, our heuristic noise-injection strategy for simulating epistemic uncertainty does not offer a strict theoretical guarantee of disentanglement from aleatoric uncertainty.  our experiments show clear empirical benefits for downstream tasks , we will revise the discussion in final version of the manuscript to make this limitation explicit.
- Regarding the baseline performance concerns on datasets with a large number of classes raised by XHmk and Vmcj, since most existing EDL baseline methods have not been evaluated on such datasets, we will release the complete implementation ensure transparency and  fair comparisons.

We sincerely thank the reviewers and AC for their careful reading and valuable suggestions, which improved both the technical quality and broader impact of our work.

---

### Decision · Program_Chairs · 2025-09-17

**Decision:**

Accept (poster)

**Comment:**

In the recent literature, EDL has been shown to be flawed in the sense of collapsing to degenerate (Dirac) predictions when trained on standard (level 0) data, unless being regularised as proposed in the original paper by Sensoy et al. Here, to overcome this problem, the authors propose the use of vicinal supervision, thereby enhancing aleatoric uncertainty estimation. In addition, they propose noise-augmented vicinal risk minimization for better epistemic uncertainty estimation. The proposed method is analysed theoretically and evaluated empirically for OOD detection and generalization on several common image datasets.

The reviewers are quite in favor of the paper, and I do agree that the combination of vicinal risk minimization (VRM) and EDL is an appealing idea, although the noise augmentation looks somewhat ad hoc to me. Moreover, the empirical evidence is quite convincing. That said, there are a few points that bother me and that the authors should take into account when finalising their paper.

For example, the authors observe that "EDL also fails to faithfully capture aleatoric uncertainty under the same empirical risk minimization principle", and they claim this observation as a new finding (Theorem 1). However, it follows directly from Theorem 3.3 in Jürgens et al. (2024). It's true that they don't say it explicitly there, but given the obvious observation that the L_1 loss is 0 for $\delta_y$, it indeed follows immediately. By the way, Jürgens et al. (2024) also state an important technical assumption, namely a universal approximation property, which is missing in the authors' theorem/proof, making it technically incorrect. Without a universal approximation property, the learner might not be able to realize the prediction of a Dirac delata function at every point x_i. Finally, and perhaps most importantly, I would argue that the claim itself is somewhat misleading, simply because it is not a defect of EDL. Exactly the same happens for a standard (first-order) neural network: Without regularisation, it will try to fit the data as much as it can, and if it has the capacity (universal approximation property), it will simply reproduce the data -- that's nothing else than standard overfitting.

A possible drawback of the proposed method is the dependency of the learner's uncertainty representation on external parameters ($\beta$), which are controlled by the user: Isn't the degree of epistemic uncertainty directly influenced by how "smooth" the first-order "vicinal" distributions are made? Don't we then have the same problem as for original EDL, namely that the uncertainty is controlled externally and hence more or less arbitrary? In EDL, it is controlled by regularising the learner, here it is controlled by "regularising the data".

Theorem 2 looks at lower bounds on the risk, not at upper bounds, which would actually be more interesting. Then, having a lower lower bound for mixing compared to the original bound is interpreted as a result in favour of mixing. This is a least a bit unusual, isn't it? After all, these are only bounds, and we even don't don't know whether they are tight.